# PDHFormer: Progressive Dual-Head Transformer for Behavioral Choice Prediction

## Abstract

Many applications require joint prediction of interdependent behavioral choices, yet existing models often treat each choice independently (e.g., through parallel prediction heads), overlooking the influence of one on the other. In this work, we propose Progressive Dual-Head Transformer (PDHFormer), a novel framework that performs two-step prediction: the model first estimates one choice and then conditions the second on this upstream estimate through an explicit head-to-head pathway. A shared encoder captures the common structure of two prediction tasks, while the dual-head module explicitly reflect cross-choice dependence. A gated residual mechanism integrated into the embedding layer and the dual-head modules further improves the training stability and the prediction performance. Extensive experiments on an urban mobility behavioral choice dataset and a real-world manufacturing dataset demonstrate that PDHFormer consistently outperforms state-of-the-art machine learning models, deep tabular models, as well as parallel-head Transformer variants across multiple metrics. Moreover, our ablation study confirms that both the proposed progressive dual-head and gated residual mechanism are key contributors to the observed gains in different prediction tasks.

## 1 Introduction

Many real-world systems require predicting interdependent choices rather than a single target. In ride-hailing, for example, a driver first decides whether to accept a request under limited information and may revisit that decision when additional details (e.g., expected fare or detour) become available (Ashkrof et al., 2022). Similar directional dependencies arise in manufacturing, where one production decision conditions the next under evolving constraints (Sharma & Gao, 2002). In these settings, choices are not merely correlated; one choice is conditionally dependent on the other, and modeling this interdependency is essential for accurate prediction.

Standard machine learning models, such as gradient-boosted trees (Prokhorenkova et al., 2018; Chen & Guestrin, 2016) and deep models including Transformers for tabular data (Huang et al., 2020; Nassar et al., 2022; Gorishniy et al., 2025; Holzmüller et al., 2024; Bonet et al., 2024; Qu et al., 2025), typically optimize either a single target or multiple targets with parallel heads. Parallelization captures shared structure but misses directional influence: the second head does not condition on the realized output or representation of the first. As a result, cross-choice dependencies remain under-modeled (Gao et al., 2022; Gu et al., 2022; Kumar et al., 2024). Explicitly modeling and predicting one target conditional on another could better capture real-world behavioral choices. This gap motivates the need for models that can jointly learn shared representations while explicitly modeling choice dependencies.

To that end, we propose PDHFormer, a Progressive Dual-Head Transformer to explicitly capture the dependency between related prediction targets through a head-to-head progressive prediction mechanism. PDHFormer couples two prediction heads through an explicit head-to-head connection while a shared encoder captures the common structure behind two predictions. A gated residual mechanism, applied at the embedding and dual-head module, regulates information flow to improve stability and accuracy. PDHFormer performs progressive prediction by allowing the output of one head to condition the other, enabling the model to explicitly exploit interdependent decision patterns. We evaluate PDHFormer on real-world datasets from urban mobility and manufacturing

and report consistent gains over state-of-the-art machine learning models, deep tabular models, and parallel-head Transformer variants across multiple metrics. Overall, our main contributions can be summarized in threefold:

- We propose PDHFormer, a novel Progressive Dual-Head Transformer that explicitly captures dependencies between related choice predictions through a head-to-head connection, while leveraging gated residuals to balance shared and task-specific signals.

- We propose a two-step progressive prediction strategy, where the output of one head conditions the prediction of the other, enabling the model to capture interdependent decision patterns. A composite loss function is designed for the two behavioral choice predictions with a regularization term of gating residuals.

- Extensive empirical evidence on two domains show that the PDHFormer outperforms strong machine learning and deep learning baselines. We also ablate on the positive impact of the head-to-head pathway and the gate residual mechanism, and provide SHAP analyses for shedding light on the predicted behavioral choices.

## 2 RELATED WORKS

**Choice Prediction** Choice prediction refers to forecasting which alternative an individual will choose from a predefined choice set based on contextual features. Applications span transportation (Shahriar et al., 2021; Wang et al., 2021; Tamim Kashifi et al., 2022), e-commerce (Chaudhuri et al., 2021; Wang et al., 2023b), healthcare (Kothinti, 2024). Traditional research typically relies on discrete choice models (Zhao et al., 2020). We note that our task is originally developed from the discrete choice modeling (DCM) framework in mobility (Ashkrof et al., 2022), and that the prediction task we study follows the data structure commonly used in behavioral choice prediction (Shahriar et al., 2021; Chaudhuri et al., 2021; Kothinti, 2024; Martín-Baos et al., 2023; Wang et al., 2023b). In our setting, each sample corresponds to one decision instance and is represented by a single contextual feature vector containing all information relevant to that instance. Our setting therefore maps the contextual features of each sample directly to a prediction of the choice.

Gradient-boosted decision trees (GBDTs), such as XGBoost (Chen & Guestrin, 2016), Hist-GBM (Guryanov, 2019), and CatBoost (Prokhorenkova et al., 2018), have widely adopted due to their strong accuracy and scalability. With the development of deep learning, an increasing number of works have explored deep learning methods for choice prediction (Wang et al., 2023a). Extending this direction, a growing number of work has focused on designing architectures specifically for tabular inputs (Huang et al., 2020; Arik & Pfister, 2021; Gorishniy et al., 2025; Bonet et al., 2024; Holzmüller et al., 2024; Qu et al., 2025), since many choice prediction tasks are naturally formulated on structured tabular features, these models also provide competitive alternatives for capturing complex interactions in decision-making data. Some approaches systematically identifying interaction effects, such as DeepHalo (Zhang et al.) for context-dependent choice prediction.

**Multi-task Prediction** Despite these advances, most existing approaches treat choice prediction as independent classification problems, overlooking the fact that behavioral decisions are often progressive and interdependent, with one decision potentially influencing or constraining others. For example, when choosing a travel mode, an early decision to take the bus instead of the metro will directly affect the subsequent decision of which route and which transfer stops to select.

In current practice, multi-task choice prediction typically uses a shared-bottom, parallel-head design (Caruana, 1997; Silver et al., 2016; Lample et al., 2022) such as Deep Task-specific Bottom Representation Networks for mitigating task interference (Liu et al., 2023) or shared-bottom neural architectures for constructing prediction intervals (Xue et al., 2024). Attention-based methods such as the Multi-Task Attention Network (Liu et al., 2019), DenseMTL (Lopes et al., 2022), and Task Relation Attention Networks (Ma & Tan, 2020) employ task-specific or cross-task attention modules to highlight relevant shared features and dynamically exchange information across tasks. Also Graph-based approaches such as GNNs based discrete choice modeling (Tomlinson & Benson, 2024) for joint classification and regression targets predict or multi-task FP-GNN framework (Ai et al., 2022) for inhibitors prediction. Besides, several studies have explored dual-branch or dual-transformer architectures in other domains (Yao et al., 2023; Han et al., 2022; Yan et al., 2023;

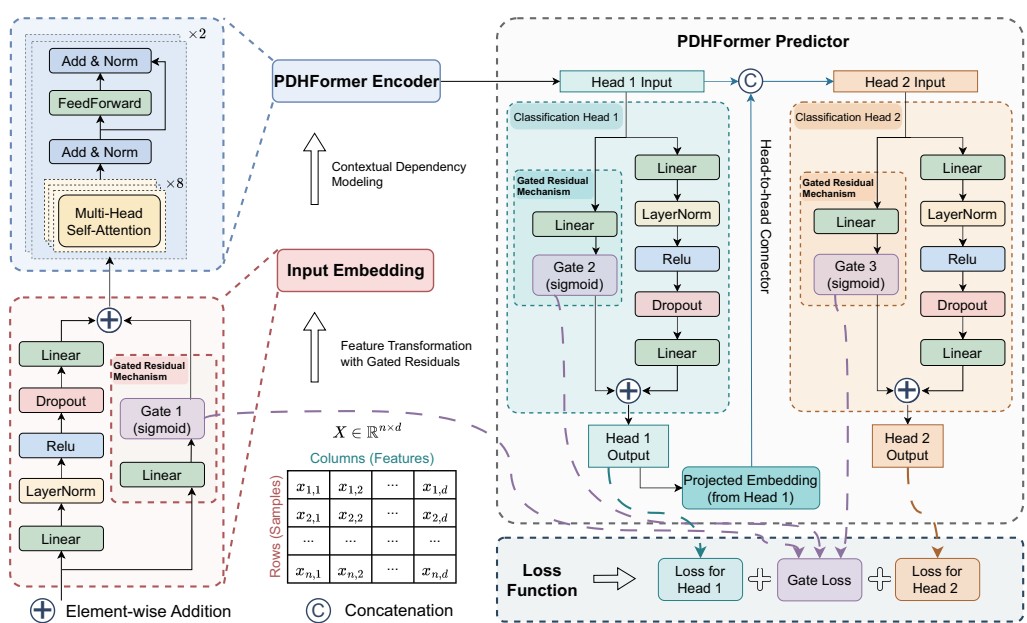

Figure 1: Overall Architecture of **PDHFormer**: A Progressive Dual-Head Transformer

Hu et al., 2022; Samoaa et al., 2024). However, these models mainly focus on multimodal fusion, 3D point clouds, medical imaging, or drug synergy prediction, and do not address within-instance dependent tabular decision modeling as studied in our work.

Given the shared representation, the outputs are effectively treated as conditionally independent, so one prediction does not inform the other. Our approach keeps the benefit of a shared encoder but introduces a progressive dual-head pathway to explicitly pass the upstream output to the downstream head, enabling the second prediction to leverage information from the first, which further improves the prediction accuracy.

## 3 PDHFORMER: A PROGRESSIVE DUAL-HEAD TRANSFORMER

Given the tabular input $X \in \mathbb{R}^{n \times d}$ with $n$ samples and $d$ features, a sample (or a record) $(\mathbf{x}_i, c_{1,i}, c_{2,i})$ is composed of features $\mathbf{x}_i \in \mathbb{R}^d$, alongside two interdependent classification targets[1] $c_{1,i} \in \{0, \ldots, C_1 - 1\}$ and $c_{2,i} \in \{0, \ldots, C_2 - 1\}$. The proposed PDHFormer maps $\mathbf{x}_i$ to a shared representation $\mathbf{h}$ using an embedding block followed by an encoder. In the predictor, we employ one classification head to predict $c_{1,i}$. Afterward, the other classification head take as input both the shared representation and the representation of $c_{1,i}$ for predicting $c_{2,i}$, through an explicit *head-to-head* connector. In doing so, the representation learned by the first head serve as an additional input to the second head, which further incorporates upstream choice information, captures its impact on the second choice, and thus improves the prediction performance. In contrast to typical parallel-head design, the PDHFormer benefits progressive predictions in real-world scenarios. For example, Choice $c_{1,\cdot}$ represents the driver's acceptance/rejection decision made under limited information, while Choice $c_{2,\cdot}$ denotes the decision with additional features such as the trip details. In some manufacturing scenarios, Choice $c_{2,\cdot}$ represents a harder regression target, which can be facilitated by first predicting a relevant classification target $c_{1,\cdot}$.

The neural architecture of the proposed PDHFormer is illustrated in Fig. 1, which mainly comprises five components:

- The input embedding layer processes the original raw features;

---

[1]Note that we use two classification targets for presenting our methodology. Our model can also be applied to the regression target, as shown in the manufacturing case in Section 4.6.

- The PDHFormer encoder advances the embeddings of features using self-attention-based layers to interact high-dimensional features and capture their complex dependencies;

- The PDHFormer predictor applies the encoder output as the shared input to the progressive dual heads, which first predict Choice $c_{1,i}$, and then predict Choice $c_{2,i}$. The representation learned from the first head is passed to second head through a head-to-head connector;

- The Gated residual mechanism is integrated into the input embedding layer and dual heads for both training stability and prediction performance improvement;

- The composite loss function is proposed to simultaneously optimize two predictions, involving a regularization term of gating residuals.

In the following, we detail the components of PDHFormer and the composite loss for jointly optimizing two predictions.

## 3.1 INPUT EMBEDDING

To enhance the representation capability of the raw input features, an embedding layer is employed to map the each feature $\mathbf{x}^j \in \mathbb{R}$ into a latent embedding. Specifically, we first apply a two-layer feedforward network, i.e., a multilayer perceptron (MLP), to project the features to high-dimensional space, in which the embeddings are regularized by layer normalization, activated by ReLU along with Dropout, and linearly transformed to dimensionality $d_{hid}$:

$$\mathbf{z}_{main} = \text{MLP}(\mathbf{x}^j) \in \mathbb{R}^{d_{hid}} \tag{1}$$

The MLP output is further added with the gated residual mechanism (see Section 3.4 for details of the gated residual mechanism). A learnable gate function, implemented as a sigmoid ($\sigma$) activated linear layer, is used to compute gate values $\mathbf{g}_{emb}$. These values are used to balance the MLP output and a residual projection of features $\mathbf{x}$, such that:

$$\tilde{\mathbf{z}} = \mathbf{g}_{emb} \odot \mathbf{z}_{main} + (1 - \mathbf{g}_{emb}) \odot \mathbf{z}_{res} \tag{2}$$

where $\odot$ denotes element-wise multiplication; the residual projection $\mathbf{z}_{res} = \mathbf{W}_{res}\mathbf{x}^j \in \mathbb{R}^{d_{hid}}$ is used to preserve original information in deep neural architecture and facilitate gradient flow.

## 3.2 PDHFORMER ENCODER

Given input embeddings $\mathbf{H}^0 = \{\tilde{\mathbf{z}}^j\}_{j=1}^d$, we feed them into a stack of encoder blocks similar to the standard architecture in (Vaswani et al., 2017). Each block is structured by a multi-head self-attention layer and a feedforward layer, to capture contextual dependencies and complex feature interactions. Specifically, we apply two identical encoder blocks, each consisting of:

**Multi-head self-attention (MHSA) layer:** Given the input embeddings $\mathbf{H}^{\ell-1}$ for the $\ell$-th block, MHSA layer is defined as:

$$\text{MHSA}\left(\mathbf{H}^{\ell-1}\right) = \text{softmax}(\frac{\mathbf{Q}_\ell \mathbf{K}_\ell^\top}{\sqrt{d_{hid}}}) \cdot \mathbf{V}_\ell, \tag{3}$$

where $\mathbf{Q}_\ell = \mathbf{W}_\ell^q \mathbf{H}^{\ell-1}$, $\mathbf{K}_\ell = \mathbf{W}_\ell^k \mathbf{H}^{\ell-1}$, and $\mathbf{V}_\ell = \mathbf{W}_\ell^v \mathbf{H}^{\ell-1}$ are the query, key, and value matrix, respectively, which are obtained from the input through linear projections.

**Feedforward network (FFN) layer:** Given the input $\mathbf{Z}^{\ell-1}$ is applied independently to each position, consisting of two linear layers with a GELU activation in between:

$$\text{FFN}(\mathbf{Z}^{\ell-1}) = \mathbf{W}_2(\text{GELU}(\mathbf{W}_1 \mathbf{Z}^{\ell-1} + \mathbf{b}_1)) + \mathbf{b}_2 \tag{4}$$

Residual connections and layer normalization in (Vaswani et al., 2017) are applied after both the MHSA and FFN layers for training stability. To sum up, the encoder updates the input embeddings $\mathbf{H}^0$ by $L$ ($L = 2$ in this work) blocks:

$$\mathbf{H}^{(L)} = \text{EncoderBlock}^{(l)}(\mathbf{H}^{(l-1)}), \quad \text{for} \quad l = 1, \dots, L \tag{5}$$

The output $\mathbf{H}^{(L)}$ serves as context-aware representations of the input features, which are further used as input to the predictor.

### 3.3 PDHFORMER PREDICTOR

PDHFormer incorporates two classification heads in the predictor for two interdependent behavioral choices, respectively. Given the context-aware representations $\mathbf{H}^{(L)} \in \mathbb{R}^{d \times d_{hid}}$ from the encoder, where $d$ is the number of features, $d_{hid}$ is the hidden dimension. We aggregate the contextual representations for each sample, such that:

$$\mathbf{h} = \frac{1}{d} \sum_{t=0}^{d-1} \mathbf{H}^{(L)}[t, :] \in \mathbb{R}^{d_{hid}} \qquad (6)$$

The classification head for the first behavioral choice $c_{1,i}$ is a feedforward network that maps $\mathbf{h}$ to a probability distribution over the categories of $c_{1,i}$. It comprises two linear layers with layer normalization, ReLU activation, and dropout. Consequently, the output dimension equals the number of categories. We also apply the gated residual mechanism that projects $\mathbf{h}$ directly to the category logits and add them to the output of the feedforward network, which are processed by Softmax for calculating categorical probabilities. The classification head for the second behavioral choice $c_{2,i}$ has the same architecture, except the output dimension equals the number of its own categories.

**Head-to-head connector:** To explicitly capture the dependency between the two behavioral choices, the output of the first classification head is incorporated into the second head via a head-to-head connector. Specifically, let $\mathbf{z}_1 \in \mathbb{R}^{C_1}$ denote the logits produced by the first head for a sample. We linearly project these logits into a low-dimensional embedding

$$\mathbf{e}_1 = \mathbf{z}_1 \mathbf{W}_c \in \mathbb{R}^{d_{\text{hid}}}, \qquad (7)$$

where $\mathbf{W}_c \in \mathbb{R}^{C_1 \times d_{\text{hid}}}$ is a learnable weight matrix. This embedding $\mathbf{e}_1$ is then concatenated with the aggregated contextual representation $\mathbf{h} \in \mathbb{R}^{d_{\text{hid}}}$ from the encoder:

$$\mathbf{h}_2 = [\mathbf{h}; \mathbf{e}_1] \in \mathbb{R}^{2d_{\text{hid}}}. \qquad (8)$$

The concatenated representation $\mathbf{h}_2$ serves as the input to the second classification head, allowing the prediction of the second choice to be explicitly conditioned on the first. In this way, the model captures interdependencies between the two behavioral choices while preserving the shared contextual information from the encoder.

### 3.4 GATED RESIDUAL MECHANISM

Formally, given the intermediate embedding $\mathbf{y}_{\text{main}}$ from the main processing pathway (i.e., the backbone) and a residual projection $\mathbf{y}_{\text{res}}$, the gate values $g$ are learned through a linear layer activated by a sigmoid function ($\sigma$):

$$g = \sigma(\mathbf{W}_g \cdot \mathbf{y}_{\text{main}} + \mathbf{b}_g) \qquad (9)$$

where $\mathbf{W}_g$ and $\mathbf{b}_g$ are trainable gate parameters. The fused output $\mathbf{y}_{\text{out}}$ is computed as a gate-value-weighted sum of $\mathbf{y}_{\text{main}}$ and $\mathbf{y}_{\text{res}}$:

$$\mathbf{y}_{\text{out}} = g \odot \mathbf{y}_{\text{main}} + (1 - g) \odot \mathbf{y}_{\text{res}} \qquad (10)$$

where $\odot$ denotes element-wise multiplication. To ensure training stability, the gate output $g$ is constrained within a reasonable range, typically clipped between 0.1 and 0.9. In this work, we further apply a regularization term that encourages the gate values to remain close to 0.5, thereby promoting a balanced integration of the backbone and residual projection rather than over-reliance on either, such that:

$$\mathcal{L}_{\text{gate}} = \mathbb{E}\left[|g - 0.5|\right] \qquad (11)$$

We integrate a gated residual mechanism into both the input embedding layer and the dual heads in the predictor. In the input embedding layer, it blends the transformed features with a projected shortcut from the raw inputs; in the heads, it combines the deeper representations learned by the head with a direct linear projection of the input. In the PDHFormer encoder, the residual connections inherently exist in the MHSA and FFN layer, without the need for additional gated residuals.

### 3.5 LOSS FUNCTION

PDHFormer is designed to predict two targets, with one of them conditionally dependent on the other. To effectively train the two interdependent classification heads while ensuring that the gated residuals are properly utilized, we employ a composite loss function that integrates the following three components, 1) $\mathcal{L}_{\text{class}}^{(c1)}$: the prediction loss for target $c_{1,\cdot}$; 2) $\mathcal{L}_{\text{class}}^{(c2)}$: the prediction loss for target $c_{2,\cdot}$; 3) $\mathcal{L}_{\text{gate}}$ (in Eq. 11): the regularization loss for the gated residual mechanism that encourages the gate values remain near 0.5.

$\mathcal{L}_{\text{class}}^{(c1)}$ and $\mathcal{L}_{\text{class}}^{(c2)}$ are implemented by the cross-entropy loss for classification. $\mathcal{L}_{\text{gate}}$ is implemented by the mean absolute error as defined by Eq. 11. Overall, the composite loss function is defined as:

$$\mathcal{L} = \alpha \mathcal{L}_{\text{class}}^{(c2)} + \beta \mathcal{L}_{\text{class}}^{(c1)} + \gamma \mathcal{L}_{\text{gate}} \tag{12}$$

where $\alpha, \beta, \gamma$ are hyperparameters controlling the relative importance of each component in the loss function. The joint optimization with the composite loss enables the PDHFormer to exploit the sequential dependency between first head and second head to improve the prediction performance.

## 4 EXPERIMENTS

### 4.1 REAL-WORLD SCENARIOS & DATASETS

We evaluate our model on two datasets: (i) an **urban mobility choice dataset** collected between November 2020 and February 2021 from Uber/Lyft drivers in the US and Uber/ViaVan drivers in the Netherlands using a simulated ride request experiment Ashkrof et al. (2022). The dataset was designed with two information sharing conditions. Under **Baseline Information Provision (BIP)**, drivers make accept/reject decisions (Choice 1) based only on limited trip attributes (e.g., request time, pickup time, rider rating) without fare or destination information. While, under **Additional Information Provision (AIP)**, drivers first make the same initial decision (Choice 1) under limited information, and are then shown additional details such as estimated fare, guaranteed tip, and traffic congestion. After receiving this enriched information, they may revise their accept/reject decision (Choice 2). and (ii) a **manufacturing dataset** consisting of high-dimensional process and production variables from an industrial environment, providing a complementary testbed to assess model robustness in complex real-world operational settings. The dataset contains two interrelated decision targets, denoted as Choice A and Choice B, which correspond to decisions governing different performance aspects of the product. In addition, Choice A is associated with a continuous performance parameters, referred to as Regression A, enabling evaluation of the model on both classification and regression objectives for the same decision aspect.

All experiments were conducted in a controlled computational environment to ensure reproducibility and consistency. The servers' hardware and software specifications are listed in Apendix E Table 9. Hyperparameters follow established practices for transformer-based tabular modeling and were finalized through extensive tuning; Apendix F details the model and training settings.

### 4.2 BASELINES & METRICS

We benchmark our model against a diverse set of baselines covering three categories: (i) classical machine learning methods, including Logistic Regression (Ng & Jordan, 2001), Naive Bayes (Murphy et al., 2006), SVMs (Joachims, 1998), Decision Trees (Song & Lu, 2015), (ii) ensemble models such as Random Forests (Breiman, 2001) and gradient-boosted decision trees (XGBoost (Chen & Guestrin, 2016), CatBoost (Prokhorenkova et al., 2018), HistGBM (Guryanov, 2019)), and (iii) recent neural architectures for tabular data, including TabTransformer (Huang et al., 2020), TabNet (Arik & Pfister, 2021), TabM (Gorishniy et al., 2025), RealMLP (Holzmüller et al., 2024), HyperFast (Bonet et al., 2024), and TabICL (Qu et al., 2025). These baselines cover both traditional and state-of-the-art approaches, ensuring a comprehensive comparison. Full implementation details are provided in Appendix H.

To comprehensively evaluate the performance of the proposed PDHFormer for urban mobility choice dataset and manufacturing dataset, we report a set of standard classification metrics, includ-

Table 1: Model comparison on the mobility dataset in the AIP scenario: for Choice 2 and Choice 1: Top 1 results are in **red**, Top 2 in yellow, and Top 3 in blue.

| Model | Choice 2 | | | | | | Choice 1 | | | | | |
|---|---|---|---|---|---|---|---|---|---|---|---|---|
| | ACC↑ | AUC↑ | AUCPR↑ | Prec↑ | Recall↑ | F1↑ | ACC↑ | AUC↑ | AUCPR↑ | Prec↑ | Recall↑ | F1↑ |
| Random Forest | 0.8382 | 0.7996 | 0.5212 | 0.7099 | 0.5995 | 0.6206 | 0.7110 | 0.7722 | **0.6329** | 0.6985 | 0.5989 | 0.5935 |
| Xgboost | 0.8353 | 0.7882 | 0.5235 | 0.7018 | 0.6583 | 0.6749 | 0.6879 | 0.7459 | 0.6043 | 0.6392 | 0.6050 | 0.6082 |
| Naive Bayes | 0.6676 | 0.6227 | 0.2574 | 0.5471 | 0.5707 | 0.4365 | 0.6214 | 0.6291 | 0.4546 | 0.5850 | 0.5892 | 0.5861 |
| Logistic Regression | 0.8179 | 0.6672 | 0.2711 | 0.5407 | 0.5065 | 0.4795 | 0.6821 | 0.6649 | 0.4901 | 0.6314 | 0.5729 | 0.5641 |
| HistGBM | 0.8295 | 0.7554 | 0.5006 | 0.6926 | 0.6683 | 0.6788 | 0.7139 | **0.7749** | 0.6298 | 0.6782 | 0.6331 | 0.6397 |
| Decision Tree | 0.7399 | 0.6278 | 0.2378 | 0.5975 | 0.6278 | 0.6054 | 0.6734 | 0.6304 | 0.4243 | 0.6322 | 0.6304 | 0.6312 |
| TabNet | 0.8179 | 0.6299 | 0.2230 | 0.4980 | 0.4998 | 0.4651 | 0.6532 | 0.5893 | 0.4134 | 0.5537 | 0.5191 | 0.4792 |
| SVM | 0.8266 | 0.7651 | 0.4369 | 0.6686 | 0.5858 | 0.6017 | 0.6965 | 0.6916 | 0.5297 | 0.6613 | 0.5880 | 0.5822 |
| TabTransformer | 0.8295 | 0.6070 | 0.2453 | 0.6880 | 0.8295 | 0.7522 | 0.6647 | 0.5054 | 0.3314 | 0.4419 | 0.6647 | 0.5309 |
| CatBoost | 0.8382 | 0.7886 | 0.5026 | 0.8194 | 0.5322 | 0.5179 | 0.7023 | 0.7311 | 0.5743 | 0.6999 | 0.5774 | 0.5595 |
| TabM | 0.8237 | 0.7848 | 0.4798 | 0.8065 | 0.8237 | 0.8131 | 0.6965 | 0.7175 | 0.5449 | 0.6784 | 0.6965 | 0.6777 |
| RealMLP | 0.8295 | 0.6131 | 0.2211 | **0.8586** | 0.8295 | 0.7522 | 0.6647 | 0.6349 | 0.4452 | **0.7771** | 0.6647 | 0.5309 |
| HyperFast | 0.8150 | 0.7833 | 0.4747 | 0.6622 | 0.6394 | 0.6489 | 0.7168 | 0.7336 | 0.5748 | 0.6788 | 0.6588 | 0.6650 |
| TabICL | 0.8410 | 0.7889 | 0.4612 | 0.7181 | 0.6147 | 0.6385 | 0.7283 | 0.7377 | 0.5793 | 0.7065 | 0.6397 | 0.6471 |
| **PDHFormer** | **0.8439** | **0.8015** | **0.5257** | 0.8205 | **0.8439** | **0.8233** | **0.7341** | 0.7579 | 0.6261 | 0.7242 | **0.7341** | **0.7120** |

ing Accuracy (ACC), Area Under the ROC Curve (AUC), Area Under the Precision–Recall Curve (AUCPR), Precision, Recall, and F1 Score. All metrics are computed on the held-out test set. For above mentioned baseline models, both Choice 2/ Choice 1 and Choice A/ Choice B predictions are evaluated separately, and results are reported individually for each task. Detailed definitions and formulae for these metrics are provided in Appendix I.

### 4.3 RESULTS FOR AIP SCENARIO: JOINT CHOICE PREDICTION

We first evaluate PDHFormer using the AIP dataset, which includes both initial driver decision (Choice 1) and the revised decision (Choice 2) made after receiving additional information. This setting allows us to assess the model's ability not only to make accurate predictions for each decision, but also to capture the dependency between them which standard parallel-head architectures are not designed to handle.

The training and validation loss curves (see Appendix A fig 3) show that the training loss steadily decreases, while the validation loss reaches its minimum around the 19th epoch. Predicted versus true labels for both Choice 2 and Choice 1 demonstrate strong alignment with ground truth, with only minor deviations for underrepresented classes (see Appendix A fig 4). This is consistent with the observed trends in Recall and F1 metrics.

Table 1 reports comparisons against baseline models. PDHFormer consistently achieves the best performance across all metrics, with substantial gains in Recall and F1, demonstrating robustness under class imbalance. To interpret model decisions, we apply SHAP analysis. Figs. 2a and 2b display the top 10 influential features for Choice 2 and Choice 1, respectively. The results show that the model captures a subset of meaningful features consistent with domain knowledge. Overall, these results demonstrate that in the AIP scenario, the proposed PDHFormer effectively predicts both Choice 2 and Choice 1.

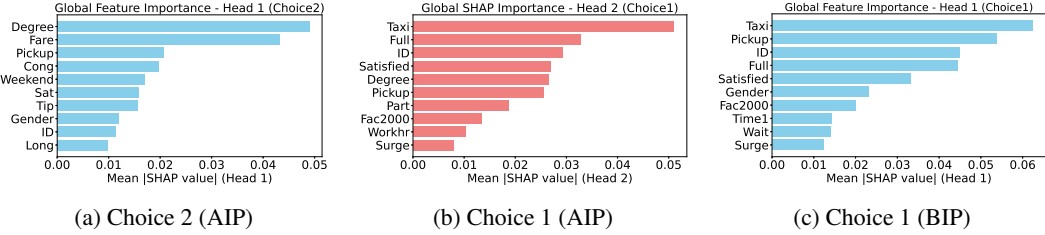

(a) Choice 2 (AIP)          (b) Choice 1 (AIP)          (c) Choice 1 (BIP)

Figure 2: Global SHAP feature importance results. Subfigures (a) and (b) correspond to the AIP scenario, while subfigure (c) shows the BIP scenario.

Table 2: Model compare for Choice 1 classification prediction in the BIP scenario: Top 1 results are in **red**, Top 2 in yellow, and Top 3 in blue. Choice 2 is **not applicable** here.

| Model | Choice 1 | | | | | |
|---|---|---|---|---|---|---|
| | ACC↑ | AUC↑ | AUCPR↑ | Prec↑ | Recall↑ | F1↑ |
| Random Forest | 0.7052 | 0.7486 | 0.5862 | 0.7256 | 0.5753 | 0.5528 |
| Xgboost | 0.6965 | 0.7512 | 0.6124 | 0.6523 | 0.6286 | 0.6340 |
| Naive Bayes | 0.6214 | 0.6328 | 0.4563 | 0.5865 | 0.5913 | 0.5876 |
| Logistic Regression | 0.6792 | 0.6654 | 0.4876 | 0.6254 | 0.5707 | 0.5620 |
| HistGBM | 0.7110 | 0.7684 | 0.6134 | 0.6767 | 0.6224 | 0.6272 |
| Decision Tree | 0.6792 | 0.6348 | 0.4289 | 0.6379 | 0.6348 | 0.6362 |
| TabNet | 0.6618 | 0.5975 | 0.4134 | 0.5816 | 0.5299 | 0.4947 |
| SVM | 0.7023 | 0.7018 | 0.5451 | 0.6753 | 0.5924 | 0.5867 |
| TabTransformer | 0.6647 | 0.5132 | 0.3544 | 0.4419 | 0.6647 | 0.5309 |
| CatBoost | 0.7023 | 0.7445 | 0.5971 | 0.6906 | 0.5817 | 0.5679 |
| TabM | 0.7052 | 0.7124 | 0.5391 | 0.6898 | 0.7052 | 0.6900 |
| RealMLP | 0.6647 | 0.6407 | 0.4805 | 0.7771 | 0.6647 | 0.5309 |
| HyperFast | 0.7139 | 0.7425 | 0.5897 | 0.6752 | 0.6502 | 0.6569 |
| TabICL | 0.7197 | 0.7196 | 0.5451 | 0.6860 | 0.6417 | 0.6493 |
| **PDHFormer** | 0.7225 | 0.6979 | 0.5465 | 0.7093 | 0.7225 | 0.7071 |

## 4.4 RESULTS FOR BIP SCENARIO: SINGLE CHOICE PREDICTION

We further evaluate the PDHFormer in the BIP scenario, where only Choice 1 prediction is required, so the second classification head is completely disabled. This experiment serves two main purposes: (i) to isolate and evaluate the impact of disabling the second head, demonstrating that the shared encoder and first classification head can still perform effectively without relying on cross-choice information; and (ii) to assess the performance of the shared encoder and the first head when the model is trained in a purely single–target setting, thereby verifying that the progressive dual–head design does not degrade performance on simpler tasks. Training and validation loss curves clearly confirm effective optimization and stable convergence throughout training (see Appendix B fig 5a). Predicted versus true labels for Choice 1 demonstrate strong alignment with the ground truth, with only minor deviations for underrepresented classes (see Appendix B fig 5b).

Table 2 compares PDHFormer with the same baseline models. Only one classification target Choice 1 is needed in the BIP scenario, and PDHFormer again achieves the best results across all metrics, with clear improvements in Recall and F1. Finally, SHAP analysis provides interpretability. Fig. 2c highlights the top 10 influential features for Choice 1. The results show that the model relies on a subset of meaningful features consistent with domain knowledge.

## 4.5 RESULTS FOR MANUFACTURING SCENARIO: JOINT CHOICE PREDICTION

To further validate the practical applicability of the proposed PDHFormer model, we conducted an exploratory evaluation on a real-world manufacturing dataset. Unlike the AIP and BIP scenarios which focus on simulated decision prediction tasks (Choice 2 and Choice 1), this dataset represents actual production-line measurements with highly imbalanced class distributions and complex feature dependencies. In this setting, we consider two categorical prediction tasks, denoted as Choice A and Choice B, which together reflect critical decision variables in the manufacturing flow. Training and validation loss curves confirm stable and effective optimization (see Appendix C fig 6). Predicted versus true labels demonstrate strong alignment with ground truth for both tasks, with minor deviations for underrepresented classes (see Appendix C fig 7 ). We follow the same experimental setup as described before. Also compared the model performance with the same baselines as before.

Due to the substantially larger size of this dataset compared to the AIP and BIP datasets, the Tab-Transformer model encountered GPU out-of-memory errors even on a device with 24 GB of memory. Its results are therefore not reported here. This observation highlights the scalability challenges faced by some Transformer-based tabular models when applied to large-scale industrial data, and further underscores the computational efficiency of our proposed Dual-Head Transformer.

Table 3 summarizes the classification performance across all models for Choice A and Choice B, respectively. Our PDHFormer consistently outperforms baselines in terms of Accuracy, Recall, and F1 score on both tasks, highlighting its ability to effectively capture complex non-linear patterns in

Table 3: Model comparison on the manufacturing dataset for Choice A and Choice B: Top 1 results are in red, Top 2 in yellow, and Top 3 in blue.

| Model | Choice A | | | | | | Choice B | | | | | |
|---|---|---|---|---|---|---|---|---|---|---|---|---|
| | ACC↑ | AUC↑ | AUCPR↑ | Prec↑ | Recall↑ | F1↑ | ACC↑ | AUC↑ | AUCPR↑ | Prec↑ | Recall↑ | F1↑ |
| Random Forest | 0.8578 | 0.9593 | 0.9492 | 0.8904 | 0.5805 | 0.6067 | 0.7454 | 0.8772 | 0.8154 | 0.6984 | 0.3233 | 0.3099 |
| XGBoost | 0.8699 | 0.9566 | 0.9457 | 0.8925 | 0.6217 | 0.6312 | 0.7797 | 0.9031 | 0.8606 | 0.7768 | 0.4639 | 0.4997 |
| Naive Bayes | 0.3959 | 0.7287 | 0.6670 | 0.3619 | 0.7004 | 0.2846 | 0.0781 | 0.5829 | 0.4917 | 0.2348 | 0.2976 | 0.0650 |
| Logistic Regression | 0.8690 | 0.9585 | 0.9422 | 0.6284 | 0.6368 | 0.6318 | 0.7528 | 0.8748 | 0.8054 | 0.6029 | 0.5620 | 0.4113 |
| HistGBM | 0.8151 | 0.8377 | 0.7621 | 0.5771 | 0.5793 | 0.5776 | 0.6413 | 0.7041 | 0.5984 | 0.3507 | 0.3600 | 0.3540 |
| Decision Tree | 0.8383 | 0.8466 | 0.7745 | 0.6085 | 0.6074 | 0.6080 | 0.7128 | 0.7514 | 0.6317 | 0.6573 | 0.4722 | 0.4641 |
| TabNet | 0.8652 | 0.9601 | 0.9442 | 0.8684 | 0.6159 | 0.6171 | 0.7723 | 0.8986 | 0.8487 | 0.8308 | 0.4653 | 0.4964 |
| SVM | 0.8299 | 0.9406 | 0.9196 | 0.8634 | 0.5735 | 0.5835 | 0.7221 | 0.8497 | 0.7697 | 0.8889 | 0.3133 | 0.3001 |
| TabTransformer | - | - | - | - | - | - | - | - | - | - | - | - |
| CatBoost | 0.8699 | 0.9632 | 0.9540 | 0.9049 | 0.5969 | 0.6207 | 0.7556 | 0.8866 | 0.8317 | 0.9024 | 0.3279 | 0.3140 |
| TabM | 0.7528 | 0.8713 | 0.8055 | 0.7507 | 0.7528 | 0.7456 | 0.7052 | 0.7124 | 0.5391 | 0.6898 | 0.7052 | 0.6900 |
| RealMLP | 0.5743 | 0.7889 | 0.7139 | 0.7555 | 0.5743 | 0.4191 | 0.4981 | 0.6171 | 0.5469 | 0.7500 | 0.4981 | 0.3313 |
| HyperFast | 0.8559 | 0.9424 | 0.9174 | 0.8635 | 0.5976 | 0.6042 | 0.7379 | 0.8568 | 0.7761 | 0.7430 | 0.3568 | 0.3750 |
| TabICL | 0.8225 | 0.9197 | 0.8988 | 0.8464 | 0.5202 | 0.5468 | 0.7314 | 0.8486 | 0.7789 | 0.8257 | 0.3328 | 0.3252 |
| **PDHFormer** | **0.8745** | 0.9593 | 0.9477 | 0.8758 | **0.8745** | **0.8746** | **0.8374** | **0.9322** | **0.8822** | 0.8342 | **0.8374** | **0.8307** |

real-world manufacturing data. These results demonstrate the model's potential for deployment in production settings to assist in decision optimization.

## 4.6 RESULTS FOR MANUFACTURING SCENARIO: CHOICE-LINKED REGRESSION

To further evaluate model performance on the manufacturing dataset, we adapt the second classification head of PDHFormer for regression prediction. Specifically, the output layer and loss function are modified to predict the continuous target *Regression A*, corresponding to Choice B in the original setup, while keeping the remaining network architecture unchanged, including the gated residual mechanism. The first classification target remains Choice A, the baseline models prediction results here are same as the table 3. The Regression A is the Choice A's associated performance parameters. To evaluate the regression prediction performance, we use Mean Absolute Error (MAE), Mean Squared Error (MSE), Root Mean Squared Error (RMSE), $R^2$ Score, and Pearson Correlation Coefficient (PCC). Besides, we noticed that GradientBoostingRegressor model is only support the regression prediction for target Regression A; TabTransformer model encountered GPU out-of-memory errors even on a device with 24 GB of memory; HyperFast model and TabICL model are only support the classification prediction for target Choice A.

Table 4 reports the regression performance of PDHFormer and baseline models on the manufacturing dataset. Across all metrics, PDHFormer consistently achieves the lowest MAE and RMSE, while attaining the highest $R^2$, demonstrating its superior capability to capture the relationship between Choice A and the associated continuous target Regression A. These results highlight that the progressive dual-head design, together with the gated residual mechanism, not only benefits classification performance but also effectively extends to regression tasks, confirming the versatility and robustness of PDHFormer in multi-target prediction scenarios.

Table 4: Model compare for regression prediction on manufacturing dataset : Top 1 results are in red, Top 2 in yellow, and Top 3 in blue

| Model | Choice A | | | | | | Regression A | | | | |
|---|---|---|---|---|---|---|---|---|---|---|---|
| | ACC↑ | AUC↑ | AUCPR↑ | Prec↑ | Recall↑ | F1↑ | MAE↓ | MSE↓ | RMSE↓ | $R^2$↑ | PCC↑ |
| Random Forest | 0.8578 | 0.9593 | 0.9492 | 0.8904 | 0.5805 | 0.6067 | 0.0061 | 0.0001 | 0.0075 | 0.7332 | 0.8699 |
| XGboost | 0.8699 | 0.9566 | 0.9457 | 0.8925 | 0.6217 | 0.6312 | 0.0059 | 0.0001 | 0.0072 | 0.7560 | 0.8890 |
| GradientBoostingRegressor | - | - | - | - | - | - | 0.0059 | 0.0001 | 0.0072 | 0.7501 | 0.8780 |
| HistGBM | 0.8151 | 0.8377 | 0.7621 | 0.5771 | 0.5793 | 0.5776 | 0.0059 | 0.0001 | 0.0072 | 0.7497 | 0.8780 |
| Decision Tree | 0.8383 | 0.8466 | 0.7745 | 0.6085 | 0.6074 | 0.6080 | 0.0059 | 0.0001 | 0.0072 | 0.7543 | 0.8691 |
| TabNet | 0.8652 | 0.9601 | 0.9442 | 0.8684 | 0.6159 | 0.6171 | 0.0060 | 0.0001 | 0.0079 | 0.7015 | 0.8378 |
| SVM/SVR | 0.8299 | 0.9406 | 0.9196 | 0.8634 | 0.5735 | 0.5835 | 0.0057 | 0.0001 | 0.0071 | 0.7565 | 0.8701 |
| TabTransformer | - | - | - | - | - | - | - | - | - | - | - |
| CatBoost | 0.8699 | 0.9632 | 0.9540 | 0.9049 | 0.5969 | 0.6207 | 0.0059 | 0.0001 | 0.0072 | 0.7512 | 0.8836 |
| TabM | 0.7528 | 0.8713 | 0.8055 | 0.7507 | 0.7528 | 0.7456 | 0.0061 | 0.0001 | 0.0078 | 0.7136 | 0.8456 |
| RealMLP | 0.5743 | 0.7889 | 0.7139 | 0.7555 | 0.5743 | 0.4191 | 0.0105 | 0.0002 | 0.0125 | 0.2540 | 0.5145 |
| HyperFast | 0.8559 | 0.9424 | 0.9174 | 0.8635 | 0.5976 | 0.6042 | - | - | - | - | - |
| TabICL | 0.8225 | 0.9197 | 0.8988 | 0.8464 | 0.5202 | 0.5468 | - | - | - | - | - |
| **PDHFormer** | **0.8745** | 0.9639 | 0.9536 | 0.8786 | **0.8745** | **0.8752** | **0.0055** | 0.0001 | **0.0071** | 0.7578 | 0.8751 |

## 4.7 ABLATION STUDY

Table 5 summarizes the impact of two core designs in PDHFormer: the gated residual (GR) mechanism and the progressive prediction structure. Our default model, **PDHFormer (Choice 2 → Choice 1)**, conditions the Choice 1 head on the hidden representation produced by the Choice 2 head. We compare it against three ablation variants: (i) **No-GR**, which removes the GR mechanism; (ii) **Parallel**, where both heads predict independently; and (iii) **Choice 1 → Choice 2**, which reverses the prediction direction.

Removing the GR mechanism consistently reduces performance. For example, in the BIP scenario, disabling GR lowers ACC from 0.7225 to 0.6994, F1 from 0.7071 to 0.6774 and Recall from 0.7225 to 0.6994, despite using the same single-head architecture. The similar drops are observed in AIP and manufacturing scenarios, indicating that GR mechanism improves learning stability and enhances feature representation across tasks, leading to more accurate predictive performance across both classification heads.

Comparing prediction structures, the progressive design (either Choice 1 → Choice 2 or Choice 2 → Choice 1) outperforms the parallel variant across all datasets. This confirms that explicitly modeling interdependence between decisions is beneficial. The direction matters: the decision predicted second consistently achieves the strongest results. In AIP, conditioning Choice 1 on Choice 2 (Choice 2 → Choice 1) gives the best Choice 1 performance, for example, Recall 0.7341 and F1 0.7120, exceeding the reversed direction. When predicting Choice 2, conditioning on Choice 1 (Choice 1 → Choice 2) gives higher scores on five of six metrics; the only exception is AUC, which is slightly higher under Choice 2 → Choice 1. The manufacturing dataset shows the same rule: predicting Choice A is best with Choice B → Choice A, and predicting Choice B is best with Choice A → Choice B; reversing the latter lowers Choice B F1 from 0.8307 to 0.7639. This pattern is expected: the second head receives an additional informative representation from the first head, thus enhancing its predictive accuracy. For the first head, its improvement (over parallel variant) might be because the shared common representation enhanced by training the second head under richer interdependent supervised signals. These comparisons demonstrate that both the gated residual mechanism and the progressive dual-head contribute substantially to the performance of PDHFormer.

Table 5: Ablation study for PDHFormer under different scenarios. Choice 1/ Choice 2 and Choice A/ Choice B corresponds to the specified input settings in each scenario. Top 1 results are in **red**.

| Model | ACC↑ | AUC↑ | AUCPR↑ | Prec↑ | Recall↑ | F1↑ | ACC↑ | AUC↑ | AUCPR↑ | Prec↑ | Recall↑ | F1↑ |
|---|---|---|---|---|---|---|---|---|---|---|---|---|
| | **AIP Scenario Choice 2** | | | | | | **AIP Scenario Choice 1** | | | | | |
| PDHFormer (No-GR) | 0.8410 | 0.7895 | 0.5182 | 0.8138 | 0.8410 | 0.8145 | 0.7168 | 0.7541 | 0.6143 | 0.7021 | 0.7168 | 0.6946 |
| PDHFormer (Parallel) | 0.8353 | 0.7867 | 0.5042 | 0.8076 | 0.8353 | 0.8124 | 0.7023 | 0.7452 | 0.5950 | 0.6852 | 0.7023 | 0.6839 |
| PDHFormer (Choice1→Choice2) | **0.8497** | 0.7953 | **0.5301** | **0.8276** | **0.8497** | **0.8257** | 0.7052 | 0.7312 | 0.5972 | 0.6907 | 0.7052 | 0.6651 |
| **PDHFormer (Choice2→Choice1)** | 0.8439 | **0.8015** | 0.5257 | 0.8205 | 0.8439 | 0.8233 | **0.7341** | **0.7579** | **0.6261** | **0.7242** | **0.7341** | **0.7120** |
| | **BIP Scenario Choice 2 (N/A)** | | | | | | **BIP Scenario Choice 1** | | | | | |
| PDHFormer (No-GR) | - | - | - | - | - | - | 0.6994 | 0.6939 | **0.5643** | 0.6808 | 0.6994 | 0.6774 |
| **PDHFormer (Ours)** | - | - | - | - | - | - | **0.7225** | **0.6979** | 0.5465 | **0.7093** | **0.7225** | **0.7071** |
| | **Manufacturing Scenario Choice A** | | | | | | **Manufacturing Scenario Choice B** | | | | | |
| PDHFormer (No-GR) | 0.8606 | 0.9519 | 0.9362 | 0.8605 | 0.8606 | 0.8604 | 0.8355 | 0.9311 | 0.8792 | 0.8328 | 0.8355 | 0.8286 |
| PDHFormer (Parallel) | 0.8652 | 0.9547 | 0.9383 | 0.8675 | 0.8652 | 0.8646 | 0.7481 | 0.8784 | 0.8171 | 0.7413 | 0.7481 | 0.7353 |
| PDHFormer (ChoiceB→ChoiceA) | **0.9266** | **0.9851** | **0.9803** | **0.9282** | **0.9266** | **0.9267** | 0.7658 | 0.8942 | 0.8382 | 0.7627 | 0.7658 | 0.7639 |
| **PDHFormer (ChoiceA→ChoiceB)** | 0.8745 | 0.9593 | 0.9477 | 0.8758 | 0.8745 | 0.8746 | **0.8374** | **0.9322** | **0.8822** | **0.8342** | **0.8374** | **0.8307** |

## 5 CONCLUSION

In this work, we proposed the PDHFormer for progressive behavioral choice prediction, designed to jointly predict two correlated categorical decision variables. The model incorporates a dual-head output structure along with gated residual mechanism. Extensive experiments on simulated behavioral choice scenarios using an urban mobility choice dataset and a real-world manufacturing dataset demonstrate that the proposed model consistently outperforms state-of-the-art baselines, including GBDT variants and recent deep tabular models, across multiple evaluation metrics. The results confirm the effectiveness of the progressive dual-head design and gated residual mechanism in improving predictive performance, which enhance the capacity to capture complex dependencies among high-dimensional input features to improve the prediction performance, particularly for the second prediction head. Overall, the PDHFormer provides a reliable and generalizable framework for behavioral choice prediction. We will extend it to more interdependent choices in future work.

## REPRODUCIBILITY STATEMENT

We have taken extensive steps to ensure the reproducibility of our results. The main paper provides a complete description of the model architecture in Sec. 3, the training objectives and composite loss in Sec. 3.5, the experimental setup in Sec. 4, and the ablation studies in Sec. 4.7. Additional training figures are provided in Appendix A, B, and C, hyperparameter configurations in Appendix F, data preprocessing details in Appendix G, baselines descriptions in Appendix H and evaluation metrics in Appendix I.

To support reproducibility, we will release the code, preprocessing scripts, configuration files, and trained checkpoints, along with instructions to regenerate the curated datasets, upon publication.

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

## A   TRAINING AND PREDICTION FIGURES FOR AIP SCENARIO

The training and validation loss curves (Fig. 3) for the AIP scenario indicate a steady decrease in training loss, while the validation loss attains its minimum at approximately the 19th epoch. In Fig. 4a and 4b The comparison between predicted and true labels for both Choice 2 and Choice 1 exhibits strong concordance with the ground truth. These observations align with the trends reported in the Recall and F1 metrics.

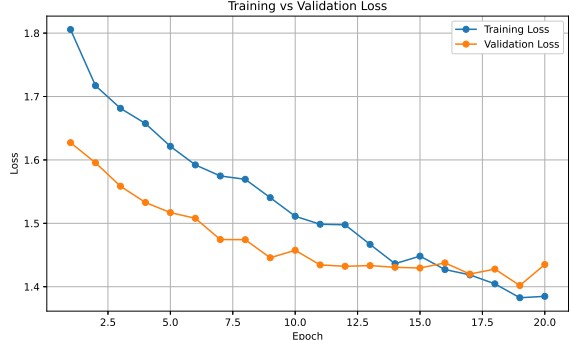

Figure 3: Training and validation loss curves of the PDHFormer model in the AIP scenario.

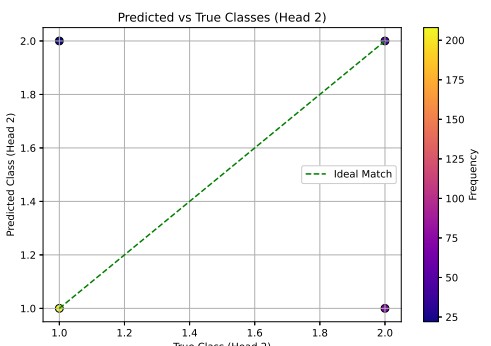
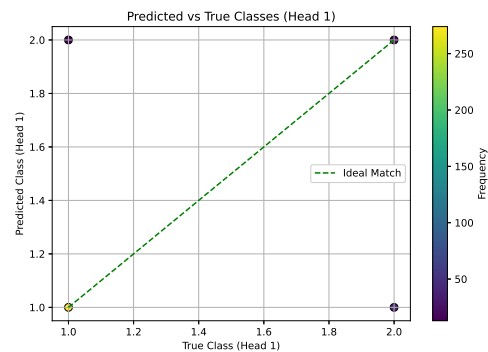

(a) Predicted vs True Labels for Choice 2 in the AIP scenario

(b) Predicted vs True Labels for Choice 1 in the AIP scenario

Figure 4: Scatter plots of predicted vs true labels on the test set.

## B TRAINING AND PREDICTION FIGURES FOR BIP SCENARIO

The training and validation loss curves for the BIP scenario (Fig. 5a) indicate that the training loss consistently decreases, while the validation loss reaches its minimum at the $9^{th}$ epoch. A slight overfitting is observed beyond this point; nevertheless, the model corresponding to the lowest validation loss is retained for evaluation. In Fig. 5b, the predicted versus true labels demonstrate good agreement with the ground truth, which is consistent with trends observed in the Recall and F1 metrics.

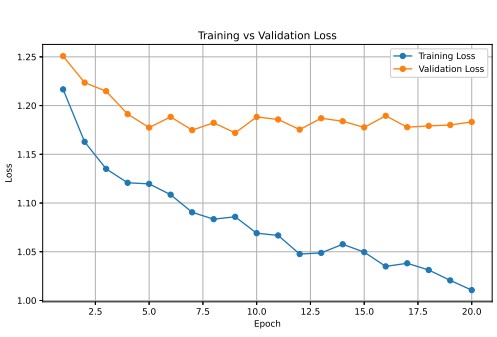
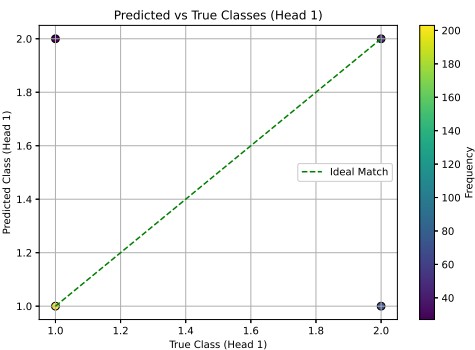

(a) Training and validation loss curves

(b) Predicted vs true labels for Choice 1 in the BIP scenario

Figure 5: BIP scenario: (a) training and validation loss curves; (b) predicted and true labels for Choice 1.

## C    TRAINING AND PREDICTION FIGURES FOR MANUFACTURING DATASET

The training and validation loss curves for the PDHFormer model on the manufacturing dataset (Fig. 6) show that the training loss steadily decreases, while the validation loss reaches its minimum at the 13[th] epoch. Slight overfitting occurs beyond this point; however, the model corresponding to the lowest validation loss is preserved for evaluation. In Fig. 7a and Fig. 7b, the predicted versus true labels align well with the ground truth, with minor deviations observed for underrepresented classes, consistent with trends in the Recall and F1 metrics.

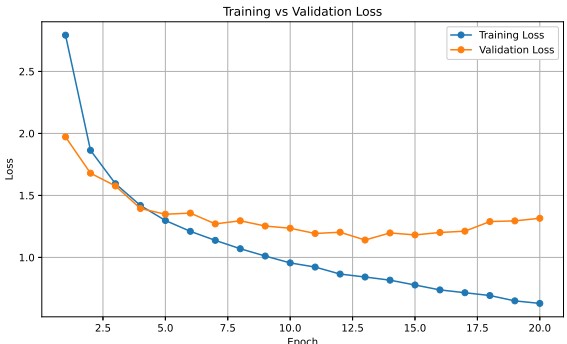

Figure 6: Training and validation loss curves of the PDHFormer model on the Manufacturing dataset.

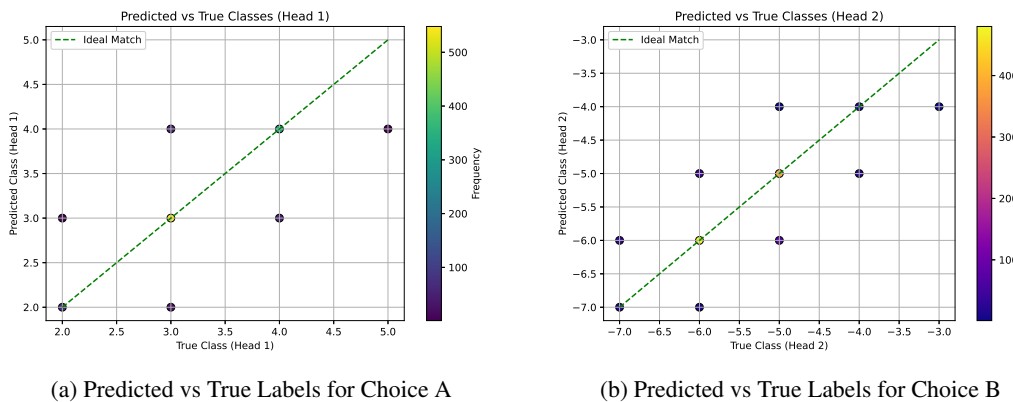

(a) Predicted vs True Labels for Choice A          (b) Predicted vs True Labels for Choice B

Figure 7: Scatter plots of predicted versus true labels for the Manufacturing dataset.

# D    EXTENDED EXPERIMENTS ON AN ADDITIONAL DELIVERY DATASET

To address reviewers' concerns regarding dataset diversity and model generalizability, we conduct additional experiments on an operational dataset derived from online delivery platform Meituan, which we refer to as the Delivery Dataset. The dataset was obtained from the Meituan INFORMS TSL Research Challenge, publicly available at: https://github.com/meituan/Meituan-INFORMS-TSL-Research-Challenge . We gratefully acknowledge that the experiment was supported by data provided by Meituan. This dataset contains three sequential decision targets, allowing us to evaluate PDHFormer under a complex multi-choice prediction setting beyond the two-choice structure presented in the main paper.

## D.1    DATASET DESCRIPTION

The dataset contains rich information describing the order dispatching and courier assignment process. Each record consists of 24 features, including:

- **Binary operational indicators**: is_courier_grabbed, is_weekend, is_prebook;
- **Geolocation coordinates**: sender and recipient locations (sender_lng, sender_lat, recipient_lng, recipient_lat) and courier location (grab_lng, grab_lat);
- **Time-related variables**: estimated arrival times, dispatching times, meal preparation times, and order push timestamps in hour–weekday–minute format.

In this dataset, we define the following three binary decision targets:

- **Head 1**: is_courier_grabbed, indicating whether a courier accepted the order;
- **Head 2**: is_weekend, reflecting whether the order occurred on a weekend day;
- **Head 3**: is_prebook, identifying whether the user requested a prebooked delivery service.

The full dataset contains 654,343 instances. To ensure efficient experimentation and fair runtime comparison against multiple baseline models, we randomly sample 10% of the data (65,434 records) while preserving the original distribution of all three decision targets.

## D.2    MODEL EXTENSION TO THE THIRD HEAD

We use the same encoder, gated residual mechanism as described in Section 3 of the main paper, and training hyperparameters as described in Appendix F, considering the instances number increase, we slightly increase the epoch number to 30.

To incorporate a third sequential decision while maintaining architectural consistency, we replicate the head-to-head connector mechanism used between the first and second heads. The replicate follows the same architectural pattern used in the two-head version as Section 3.3 of the main paper. So we refer this model as 3Head-PDHformer. Let $\mathbf{z}_2 \in \mathbb{R}^{C_2}$ denote the logits produced by the second head. We first map these logits into a low-dimensional embedding via a learnable projection:

$$\mathbf{e}_2 = \mathbf{z}_2 \mathbf{W}_c^{(2)} \in \mathbb{R}^{d_{\text{hid}}}, \tag{13}$$

where $\mathbf{W}_c^{(2)} \in \mathbb{R}^{C_2 \times d_{\text{hid}}}$ is the projection matrix for the second head.

Next, this embedding is concatenated with the shared contextual representation $\mathbf{h} \in \mathbb{R}^{d_{\text{hid}}}$ from the encoder:

$$\mathbf{h}_3 = [\mathbf{h}; \mathbf{e}_2] \in \mathbb{R}^{2d_{\text{hid}}}. \tag{14}$$

The resulting vector $\mathbf{h}_3$ serves as the input to the third prediction head, enabling the model to represent the dependency

$$\text{Choice A} \rightarrow \text{Choice B} \rightarrow \text{Choice C}.$$

This design is modular and compositional: the head-to-head connector directly mirrors the structure used in the two-head case and can be repeatedly applied, enabling PDHFormer to scale naturally to decision sequences of arbitrary length (e.g., three, four, or more interdependent choices) while preserving directional-dependency modeling and maintaining computational simplicity.

## D.3 TRAINING AND PREDICTION FIGURES

The training and validation loss curves as Fig. 8 for the delivery dataset indicate a steady decrease in training loss, while the validation loss attains its minimum at approximately the 30$^{th}$ epoch. In Fig. 9a,9b and 9c The comparison between predicted and true labels for Head 1, Head 2 and Head 3 exhibits strong concordance with the ground truth.

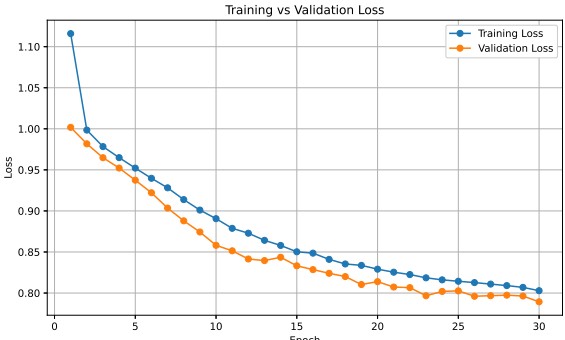

Figure 8: Training and validation loss curves of the 3Head-PDHFormer model on the Delivery dataset.

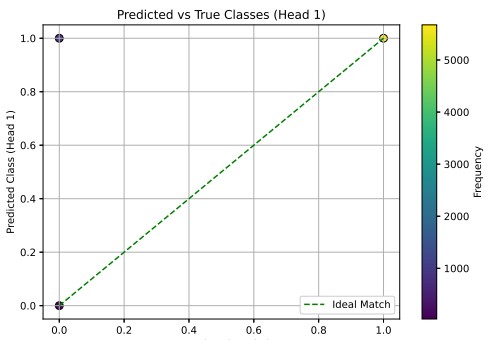

(a) Predicted vs True Labels for Head 1 (`is_courier_grabbed`) on the Delivery dataset

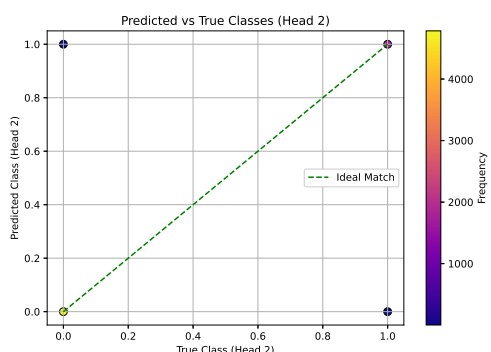

(b) Predicted vs True Labels for Head 2 (`is_weekend`) on the Delivery dataset

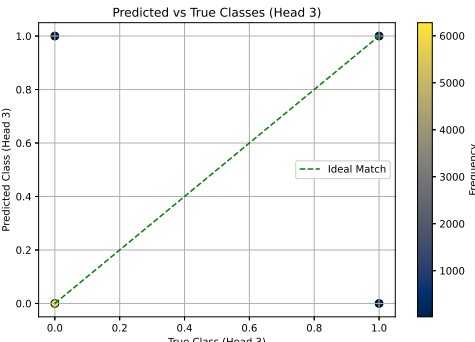

(c) Predicted vs True Labels for Head 3 (`is_prebook`) on the Delivery dataset

Figure 9: Scatter plots of predicted vs true labels on the test set.

To interpret model decisions, we apply SHAP analysis. Figs. 10a, 10b and 10c display the top 10 influential features for for Head 1, Head 2 and Head 3 respectively. The results show that the model captures a subset of meaningful features consistent with domain knowledge. Overall, these results demonstrate that for delivery dataset, the proposed model effectively predicts multiple choice.

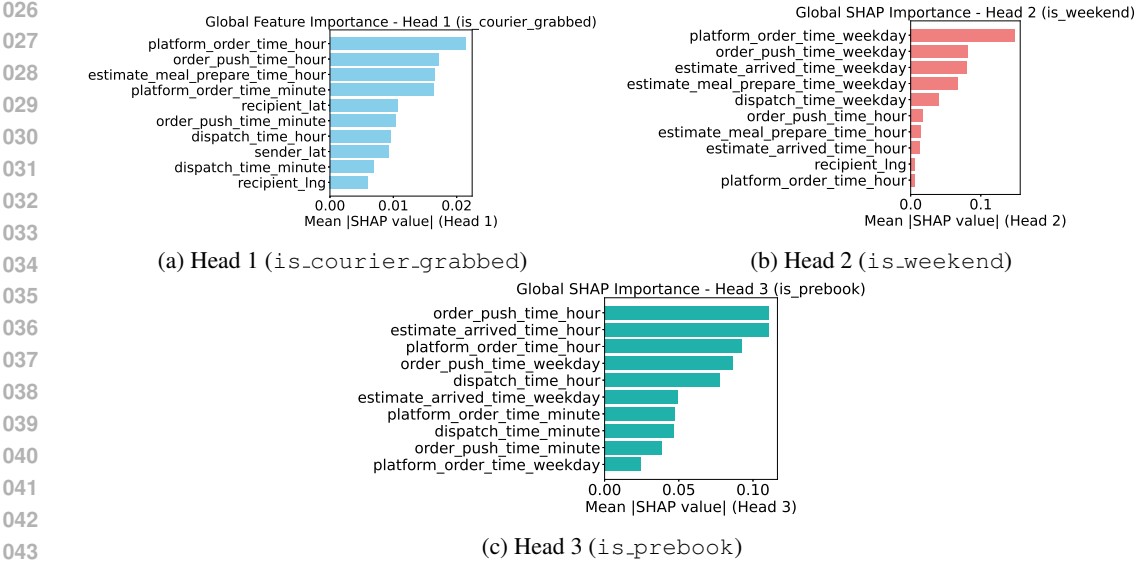

(a) Head 1 (is_courier_grabbed)

(b) Head 2 (is_weekend)

(c) Head 3 (is_prebook)

Figure 10: Global SHAP feature importance results for the delivery dataset. Subfigures (a), (b), and (c) correspond to the three prediction targets: is_courier_grabbed, is_weekend, and is_prebook, respectively.

## D.4 BASELINES COMPARE

All baseline models were extended to output three predictions accordingly. Due to time and computational constraints, we selected a subset of baselines of main paper that either performed strongly in the previous experiments or represent recent state-of-the-art approaches, and adapted them for three-choice prediction.

For evaluation, we report a set of standard classification metrics, including Accuracy (ACC), Area Under the ROC Curve (AUC), Area Under the Precision–Recall Curve (AUCPR), Precision, Recall, and F1 Score for each decision target.

Table 6, Table 7 and Table 8 summarizes the classification performance across all models for the three prediction targets: is_courier_grabbed, is_weekend, and is_prebook, respectively. Our 3Head-PDHFormer consistently outperforms baselines in terms of Accuracy, Recall, and F1 score on different tasks, highlighting its ability to effectively capture complex non-linear patterns in real-world delivery data.

Table 6: Model comparison on the delivery dataset for is_courier_grabbed: Top 1 results are in red, Top 2 in yellow, and Top 3 in blue.

| Model | ACC ↑ | AUC ↑ | AUCPR ↑ | Precision ↑ | Recall ↑ | F1 ↑ |
|---|---|---|---|---|---|---|
| Random Forest | 0.8710 | 0.7059 | 0.9368 | 0.9353 | 0.5149 | 0.4944 |
| Xgboost | 0.8671 | 0.7404 | 0.9475 | 0.9335 | 0.5000 | 0.4644 |
| CatBoost | 0.8671 | 0.7090 | 0.9394 | 0.9335 | 0.5000 | 0.4644 |
| RealMLP | 0.8671 | 0.5529 | 0.8943 | 0.8847 | 0.8671 | 0.8053 |
| HyperFast | 0.8703 | 0.7023 | 0.9391 | 0.8444 | 0.5150 | 0.4951 |
| TabICL | 0.8721 | 0.5427 | 0.8770 | 0.7969 | 0.5292 | 0.5227 |
| **3Head-PDHFormer** | 0.8710 | 0.7471 | 0.9504 | 0.8877 | 0.8710 | 0.8147 |

Table 7: Model comparison on the delivery dataset for `is_weekend`: Top 1 results are in red, Top 2 in yellow, and Top 3 in blue.

| Model | ACC ↑ | AUC ↑ | AUCPR ↑ | Precision ↑ | Recall ↑ | F1 ↑ |
|---|---|---|---|---|---|---|
| Random Forest | 0.9957 | 0.9999 | 0.9996 | 0.9965 | 0.9926 | 0.9945 |
| Xgboost | 0.9986 | 1.0000 | 1.0000 | 0.9991 | 0.9974 | 0.9982 |
| CatBoost | 0.9985 | 1.0000 | 0.999 | 0.9984 | 0.9977 | 0.9981 |
| RealMLP | 0.9833 | 0.9923 | 0.9769 | 0.9833 | 0.9833 | 0.9833 |
| HyperFast | 0.9956 | 0.9999 | 0.9997 | 0.9941 | 0.9946 | 0.9944 |
| TabICL | 0.9995 | 0.9997 | 0.9996 | 0.9997 | 0.9991 | 0.9994 |
| **3Head-PDHFormer** | 0.9989 | 1.0000 | 1.0000 | 0.9989 | 0.9989 | 0.9989 |

Table 8: Model comparison on the delivery dataset for `is_prebook`: Top 1 results are in red, Top 2 in yellow, and Top 3 in blue.

| Model | ACC ↑ | AUC ↑ | AUCPR ↑ | Precision ↑ | Recall ↑ | F1 ↑ |
|---|---|---|---|---|---|---|
| Random Forest | 0.9645 | 0.8613 | 0.5032 | 0.9822 | 0.5304 | 0.5482 |
| Xgboost | 0.9623 | 0.9434 | 0.7242 | 0.9811 | 0.5000 | 0.4904 |
| CatBoost | 0.9623 | 0.9058 | 0.5017 | 0.9811 | 0.5000 | 0.4904 |
| RealMLP | 0.9623 | 0.6377 | 0.0644 | 0.9637 | 0.9623 | 0.9437 |
| HyperFast | 0.9705 | 0.9609 | 0.7173 | 0.9763 | 0.6113 | 0.6740 |
| TabICL | 0.9914 | 0.9561 | 0.8611 | 0.9573 | 0.9216 | 0.9387 |
| **3Head-PDHFormer** | 0.9861 | 0.9874 | 0.8988 | 0.9854 | 0.9861 | 0.9853 |

# E HARDWARE INFORMATION

Table 9: Experimental Environment Specifications

| Category | Details |
|---|---|
| **Hardware Configuration** | |
| CPU | AMD Ryzen™ 9 5950X |
| GPU | NVIDIA RTX 3090 |
| RAM | 128 GB |
| Storage | NVMe SSD |
| **Software Environment** | |
| Operating System | Windows 10 |
| Python Version | 3.10.18 |
| PyTorch-GPU Version | 2.4.1 |
| CUDA Version | 12.4 |
| Scikit-learn Version | 1.26.4 |
| Pandas Version | 2.3.2 |
| Numpy Version | 1.7.1 |
| XGBoost Version | 3.0.5 |
| Matplotlib Version | 3.10.5 |
| Seaborn Version | 0.13.2 |
| SciPy Version | 1.15.3 |
| SHAP Version | 0.48.0 |

# F   HYPERPARAMETERS SETTINGS

Table 10: Hyperparameters

| Category | Details |
|---|---|
| **Model Hyperparameters** | |
| Input Embedding's Input Dimension | Features Nums |
| Input Embedding Hidden Size | 256 |
| Input Embedding's Output Dimension | 128 |
| PDHFormer Encoder Input Dimension | 128 |
| PDHFormer Encoder Hidden Dimension | 512 |
| PDHFormer Encoder Output Dimension | 128 |
| Encoder Block | 2 |
| Number of Attention Heads | 8 |
| Head 1 Input Dimension | 128 |
| Head 1 Hidden Dimension | 32 |
| Head 1 output Dimension | Class Nums (Choice 2) |
| Projected Embedding Dimension | 8 |
| Head to Head Connector Dimension Operation | 128(Head 1 Input) + 8 (Projected Embedding) |
| Head 2 Input Dimension | 136 |
| Head 2 Hidden Dimension | 32 |
| Head 2 output Dimension | Class Nums (Choice 1) |
| **Training Hyperparameters** | |
| Dropout Rate | 0.4 |
| Optimizer | AdamW |
| Learning Rate | $1 \times 10^{-4}$ |
| LR Weight Decay | $1 \times 10^{-2}$ |
| Batch Size | 32 |
| Epochs | 20 |
| Classification Loss Functions | CrossEntropyLoss |
| Classification Loss Weight $\alpha$ | 2 (Choice 2) |
| Classification Loss Weight $\beta$ | 1.2 (Choice 1) |
| Gate Regularization Loss Weight $\gamma$ | 0.01 |
| Gate Weight (Embed Stage 1) | 1 |
| Gate Weight (Classifier) | 1 |
| Gate Weight (Regressor) | 1 |
| Gate Initial Bias | 0 |
| Gate Initial Weight | Xavier uniform |
| Gate Initial Gain | 1 |
| Random Seed | 42 |
| PyTorch Deterministic Mode | True |
| PyTorch Benchmarking | False |

## G  DATA PREPROCESSING

In this paper, we evaluate on two datasets:

(i) the *Urban Mobility Choice* dataset, which can be used directly without any preprocessing.

(ii) the *Manufacturing* dataset that requires several preprocessing steps, so we implement the following steps to prepare the dataset: Empirical tests showed this threshold effectively eliminates samples with missing values; Duplicate columns and columns with near-zero variance are excluded, unless marked as important for the task; Highly correlated features (absolute Pearson correlation coefficient greater than 0.99) are removed to reduce redundancy and multicellularity; Categorical features are encoded with one-hot encoding if they have fewer than 10 unique categories; otherwise, label encoding is applied; String-type columns with constant values are dropped since they do not provide discriminative information; Boolean-type features are converted to integer format to maintain numerical consistency; All remaining features are standardized using Z-score normalization (mean = 0, standard deviation = 1), which is essential for stable convergence in neural network-based models; The calibration code as classification target is mapped linearly to integer classes starting from 0 up to the total number of unique codes, facilitating classification; The calibration value as regression target is independently normalized using a separate Z-score scaler; After feature standardization, any samples containing NaN values are removed to ensure clean inputs; The cleaned dataset is randomly split into training (80%), validation (10%), and test (10%) subsets. Stratified sampling is used based on the classification target to preserve class distribution balance; Finally, all subsets are converted to PyTorch tensors and packaged into `DataLoaders` for efficient batch-wise training and evaluation.

# H  BASELINES DETAILS

## H.1  CLASSICAL MACHINE LEARNING METHODS

- **Logistic Regression** (Ng & Jordan, 2001): A linear classifier modeling the log-odds of class probabilities, widely used for interpretable classification tasks.
- **Naive Bayes** (Murphy et al., 2006): A probabilistic model assuming conditional independence among features given the class, efficient for text and tabular classification.
- **Support Vector Machines (SVMs)** (Joachims, 1998): A margin-based classifier that learns separating hyperplanes in feature space, often effective with kernel methods.
- **Decision Trees** (Song & Lu, 2015): A non-parametric model that partitions the input space into regions using recursive feature splits.

## H.2  ENSEMBLE MODELS

- **Random Forests** Breiman (2001): An ensemble of decision trees trained on bootstrapped samples with feature randomness, reducing variance and improving generalization.
- **XGBoost** (Chen & Guestrin, 2016): A gradient-boosted decision tree algorithm optimized for speed and regularization.
- **CatBoost** (Prokhorenkova et al., 2018): A boosting method designed to handle categorical variables efficiently with ordered boosting to avoid target leakage.
- **HistGBM** (Guryanov, 2019): A histogram-based gradient boosting method improving training efficiency on large datasets.

## H.3  NEURAL ARCHITECTURES FOR TABULAR DATA

- **TabTransformer** (Huang et al., 2020): An attention-based architecture that models dependencies among categorical features via Transformer layers, enabling improved representation learning for tabular data.
- **TabNet** (Arik & Pfister, 2021): A deep tabular architecture that employs sequential attention to select salient features at each decision step, enabling both interpretability and efficient representation learning, with support for self-supervised pretraining.
- **TabM** (Gorishniy et al., 2025): A parameter-efficient ensembling approach where a single MLP imitates an ensemble of multiple MLPs by sharing most parameters, achieving strong performance and efficiency on tabular learning benchmarks.
- **RealMLP** (Holzmüller et al., 2024): Proposes optimized MLP-based architectures and training strategies for competitive performance on tabular data.
- **HyperFast** (Bonet et al., 2024): A meta-trained hypernetwork that generates dataset-specific neural networks for tabular classification in a single forward pass.
- **TabICL** (Qu et al., 2025): A tabular foundation model leveraging in-context learning, pretrained on synthetic datasets up to 60K samples; it introduces a column-then-row attention mechanism to scale ICL to large tables.

## I  EVALUATION METRIC DETAILS

Here are the classification evaluation metrics which used in our Table 1, Table 2 and Table 3. Let $y_i$ and $\hat{y}_i$ denote the true and predicted class labels, and let $C$ be the total number of classes. Here, $TP_c$ (true positives) is the number of correctly predicted samples for class $c$; $FP_c$ (false positives) is the number of samples incorrectly predicted as class $c$; $FN_c$ (false negatives) is the number of samples of class $c$ incorrectly predicted as other classes; $N_c$ is the number of true samples belonging to class $c$; and $N = \sum_{c=1}^{C} N_c$ is the total number of samples.

- **Accuracy (ACC)** measures the proportion of correctly predicted labels:

$$\text{ACC} = \frac{\sum_{c=1}^{C} TP_c}{N} \tag{15}$$

- **Area Under the Receiver Operating Characteristic Curve (AUC)**: Let $TPR(t)$ and $FPR(t)$ denote the true positive rate and false positive rate at threshold $t$, respectively. Then the AUC is computed as the integral over all thresholds:

$$\text{AUC} = \int_0^1 TPR(FPR)\, d(FPR) \tag{16}$$

- **Area Under the Precision-Recall Curve (AUCPR)**: Let Precision$(r)$ denote precision as a function of recall $r$. Then the AUCPR is:

$$\text{AUCPR} = \int_0^1 \text{Precision}(r)\, dr \tag{17}$$

- **Precision** is the average precision across classes, weighted by support:

$$\text{Precision} = \sum_{c=1}^{C} \frac{N_c}{N} \cdot \frac{TP_c}{TP_c + FP_c} \tag{18}$$

- **Recall** is the average recall across classes, also weighted:

$$\text{Recall} = \sum_{c=1}^{C} \frac{N_c}{N} \cdot \frac{TP_c}{TP_c + FN_c} \tag{19}$$

- **F1 Score** is the harmonic mean of precision and recall:

$$\text{F1} = \sum_{c=1}^{C} \frac{N_c}{N} \cdot \frac{2 \cdot \text{Precision}_c \cdot \text{Recall}_c}{\text{Precision}_c + \text{Recall}_c} \tag{20}$$

Except for the common classification metrics, we use Mean Absolute Error (MAE), Mean Squared Error (MSE), Root Mean Squared Error (RMSE), $R^2$ Score, and Pearson Correlation Coefficient (PCC) to evaluate the regression prediction performance in Appendix **??** Table 4. Let $y_i$ and $\hat{y}_i$ denote the true and predicted regression values.

- **Mean Absolute Error (MAE)**:

$$\text{MAE} = \frac{1}{N} \sum_{i=1}^{N} |\hat{y}_i - y_i| \tag{21}$$

- **Mean Squared Error (MSE)**:

$$\text{MSE} = \frac{1}{N} \sum_{i=1}^{N} (\hat{y}_i - y_i)^2 \tag{22}$$

- **Root Mean Squared Error (RMSE)**:

$$\text{RMSE} = \sqrt{\text{MSE}} \tag{23}$$

- $R^2$ **Score**:

$$R^2 = 1 - \frac{\sum_{i=1}^{N}(\hat{y}_i - y_i)^2}{\sum_{i=1}^{N}(y_i - \bar{y})^2} \tag{24}$$

where $\bar{y}$ denotes the mean of the true values.

- **Pearson Correlation Coefficient (PCC)**:

$$\text{PCC} = \frac{\sum_{i=1}^{N}(\hat{y}_i - \bar{\hat{y}})(y_i - \bar{y})}{\sqrt{\sum_{i=1}^{N}(\hat{y}_i - \bar{\hat{y}})^2} \cdot \sqrt{\sum_{i=1}^{N}(y_i - \bar{y})^2}} \tag{25}$$

All metrics are computed on the held-out test set. Weighted classification metrics are used to account for class imbalance, and inverse scaling is applied to regression outputs for correct unit interpretation.

## J  DATASET FEATURE LISTS

This appendix provides the complete feature lists for the two publicly describable datasets used in our experiments: (1) the *Urban Mobility Choice Dataset* under AIP and BIP scenarios, and (2) the *Delivery Dataset* derived from an online delivery platform. For reproducibility and clarity, we list all features for these two datasets in Tables 11 and 12, respectively.

The Urban Mobility Choice Dataset contains behavioral, temporal, demographic, and contextual features used in the AIP/BIP choice prediction tasks. The complete list of features is summarized in Table 11.

Table 11: Complete feature list of the Urban Mobility Choice Dataset used in AIP/BIP scenarios.

| Category | Features |
|---|---|
| All Features | ID, Choice1, Choice2, Req, Time, Time1, Time2, Wait, Dec, Rate, Pickup, Loc, Surge, Long, Cong, Tip, Fare, Block, Workhr, Part, Full, Age, Beginners, Experienced, Acceptance, EarnInc, ExpInc, Satisfied, Taxi, Gender, Partner, Degree, NY, CA, NY_CA, Morning, Midday, Afternoon, Evening, Night, Fac1000, Fac2000, Weekend, Weekend_Friday, Sat_Fri, Sat, Thu_Fri_Sat, Peak_evening, Peak_morning, Peak |

The delivery dataset includes temporal, spatial, and operational features from an online delivery platform. These variables relate to three sequential decision targets: `is_courier_grabbed`, `is_weekend`, and `is_prebook`. All features are listed in Table 12.

Table 12: Complete feature list for the Delivery Dataset.

| Category | Features |
|---|---|
| All Features | is_courier_grabbed, is_weekend, is_prebook, sender_lng, sender_lat, recipient_lng, recipient_lat, grab_lng, grab_lat, estimate_arrived_time_hour, estimate_arrived_time_weekday, estimate_arrived_time_minute, dispatch_time_hour, dispatch_time_weekday, dispatch_time_minute, estimate_meal_prepare_time_hour, estimate_meal_prepare_time_weekday, estimate_meal_prepare_time_minute, order_push_time_hour, order_push_time_weekday, order_push_time_minute, platform_order_time_hour, platform_order_time_weekday, platform_order_time_minute |

Another dataset used in the main paper, the *manufacturing scenario dataset* is derived from an industrial production line. Due to confidential, we are unable to disclose the full list of feature names. However, all experimental procedures, model configurations, and evaluation protocols remain fully documented to ensure scientific transparency.

# K    COMPUTATIONAL COST COMPARISON

Following the reviewer's suggestion to report computational cost, we extend our architecture by introducing a parallel dual-encoder variant, consistent with prior dual-path Transformer designs (Yao et al., 2023; Han et al., 2022; Yan et al., 2023; Hu et al., 2022; Samoaa et al., 2024). Unlike "parallel-head" designs, these works process the same input through two separate Transformer encoders and merge the representations at the latent level. To compare with this line of work, we modify PDHFormer as follows:

- The first encoder processes the embedded input:

$$x_{\text{enc1}} = \text{TransformerEncoder}_1(x_{\text{embed}})$$

- The second encoder processes the same embedded input independently:

$$x_{\text{enc2}} = \text{TransformerEncoder}_2(x_{\text{embed}})$$

- The outputs are averaged:

$$x_{\text{enc}} = 0.5 \cdot (x_{\text{enc1}} + x_{\text{enc2}})$$

The aggregated representation is then fed into the PDHFormer predictor. We compare the computational cost and predictive performance of the original **single-encoder PDHFormer** and the new **parallel-encoder PDHFormer**. Experiments are conducted on the Delivery Dataset under the 3-choice setting for 30 epochs. Table 13 compared the training and inference cost, and Table 14 compared the single-encoder and parallel-encoder performance.

Table 13: Single-Encoder and Parallel-Encoder Computational Cost Comparison

| Metric / Model Variant | Single-Encoder PDFormer | Parallel-Encoder PDFormer |
|---|---|---|
| Training Time (s) | 394.85 | 460.77 |
| Inference Time (per batch, ms) | 1.67 | 2.42 |
| Inference Time (per sample, ms) | 0.0260 | 0.0378 |

Table 14: Single-Encoder and Parallel-Encoder Performance Comparison Across All Heads

| Head | Metric | Single-Encoder | Parallel-Encoder |
|---|---|---|---|
| Head 1: is_courier_grabbed | ACC ↑ | 87.10% | 87.09% |
| | AUC ↑ | 0.7471 | 0.7429 |
| | AUCPR ↑ | 0.9504 | 0.9498 |
| | Precision ↑ | 0.8877 | 0.8786 |
| | Recall ↑ | 0.8710 | 0.8709 |
| | F1 ↑ | 0.8147 | 0.8149 |
| Head 2: is_weekend | ACC ↑ | 99.89% | 99.94% |
| | AUC ↑ | 1.0000 | 1.0000 |
| | AUCPR ↑ | 1.0000 | 1.0000 |
| | Precision ↑ | 0.9989 | 0.9994 |
| | Recall ↑ | 0.9989 | 0.9994 |
| | F1 ↑ | 0.9989 | 0.9994 |
| Head 3: is_prebook | ACC ↑ | 98.61% | 98.52% |
| | AUC ↑ | 0.9874 | 0.9867 |
| | AUCPR ↑ | 0.8988 | 0.8826 |
| | Precision ↑ | 0.9854 | 0.9844 |
| | Recall ↑ | 0.9861 | 0.9852 |
| | F1 ↑ | 0.9853 | 0.9842 |

The parallel dual-encoder design substantially increases computational cost while offering negligible performance improvement. Training and inference times increase by roughly 45%, while accuracy, AUC, and F1 vary by less than 0.1%. This confirms that the single-encoder PDHFormer is more efficient, justifying the use of the single-encoder architecture in the main paper.

## L    THE USE OF LARGE LANGUAGE MODELS (LLMS)

In this work, large language models were leveraged to assist in refining and formatting LaTeX content, including tables, figures, and equations. The LLMs provided suggestions for improving clarity, consistency, and alignment of visual elements, ensuring that all figures and tables adhered to publication-quality standards while reducing manual editing effort.

