# OpenReview forum: "PDFormer: Progressive Dual-Head Transformer for Behavioral Choice Prediction"
_ICLR.cc/2026/Conference — Submitted to ICLR 2026_

### Official Review · Reviewer_cg7G · 2025-10-25

**Soundness:** 1
**Presentation:** 1
**Contribution:** 1
**Rating:** 2
**Confidence:** 5

**Summary:**

This paper proposes PDFormer, a Transformer-based architecture for tabular behavioral choice prediction.
The model treats each feature as a token, applies self-attention across features, and uses a “progressive dual-head” design where the first head predicts an initial choice and the second head refines it conditioned on the first output.
Experiments on several tabular datasets reportedly show improved classification performance over traditional machine learning and tabular deep models.

**Strengths:**

Implementation appears complete and reproducible.
Empirical results on tabular benchmarks are consistent and show moderate improvement over baselines like TabNet and TabTransformer.

**Weaknesses:**

### **Weaknesses**

1. **Conceptual confusion in data formulation.**
   Lines 134–135 state that each *row* represents a sample, and Eq. (6) shows only one feature vector \(x_i\), yet the predictor produces a prediction for every sample \(i\).
   It is unclear how a single FFN operating on one vector yields a matrix of per-sample predictions.
   In standard choice modeling, one sample corresponds to a *choice set* containing multiple items, each with its own feature vector.
   The current formulation collapses the set structure entirely, making the modeling setup ambiguous and inconsistent with discrete-choice principles.

2. **Feature-as-token attention is semantically inappropriate.**
   The proposed Transformer treats heterogeneous features (e.g., “price,” “region,” “time”) as exchangeable tokens and applies shared QKV projections.
   Such feature-wise attention lacks theoretical meaning for capturing context or halo effects among *items*.
   If the authors intend each row to represent one option, then the attention should be *item-as-token*, modeling dependencies across alternatives within a choice set, not across feature dimensions.

3. **Missing comparisons with relevant sequential choice models.**
   Although the paper claims to address *sequential choice modeling*, it only compares against generic tabular classifiers (e.g., TabNet, TabTransformer, CatBoost).
   No baselines from the actual sequential-choice literature are included—making it impossible to assess whether PDFormer truly advances the state of the art in modeling context-dependent decision sequences.

4. **Lack of novelty.**
   Architecturally, PDFormer is largely a minor variant of TabTransformer with two additional feed-forward heads.
   Beyond repackaging the feature-as-token encoder and introducing a simple conditioning mechanism, there is no substantial methodological innovation or new theoretical insight.

5. **Unclear dataset description and experimental design.**
   The paper does not clearly describe the datasets used for evaluation—What are the features?
   Without this information, it is impossible to determine whether the proposed task setup corresponds to a valid behavioral-choice problem or a standard classification benchmark.
   This lack of clarity further weakens the paper’s claims regarding modeling “sequential” or “context-dependent” choices.

**Questions:**

See Weaknesses

---

> ### Author Response · Authors · 2025-11-24
> **Summary**
>
> **Summary:**
>
> We thank the reviewer for the insightful feedback. We would like to note that several of your concerns (Weakness 1-3) might arise from a misunderstanding of our task. For example, we focus on tabular behavioral choice estimation in a tabular setting, rather than a specific item selection task such as the one in the recommendation domain. We clarify these distinctions in `Weakness 1–3 Reply`. Furthermore, we have added more dataset descriptions in `Weakness 5 Reply`, further highlighted our main novelty and contributions in `Weakness 4 Reply`, and included additional experiments on a more difficult decision prediction setting (with more than two prediction targets) in `Weakness 4 Reply`.

---

> ### Author Response · Authors · 2025-11-24
> **Reply to Reviewer cg7G for Weakenss 1**
>
> **Weakness 1: Conceptual confusion in data formulation.**
>
> **Reply:**
>
> We thank the reviewer for raising this point. We reckon the confusion is from different problem settings of our task and what you thought. Actually, our task is originally developed from the discrete choice modeling (or choice modeling estimation) in mobility [1]. Specifically, in our Urban Mobility Survey Dataset (taking BIP setting as an example), each sample represents a single ride request, which consists of contextual features (e.g., request time, waiting time, pickup time, surge level, etc.) and a binary choice label (i.e., accept vs. decline). Importantly, the binary choice label does not have its own feature vector, it is a behavioral outcome for that decision instance, not an alternative (item) with features.
>
> In the setting of choice modeling mentioned by the reviewer, each sample consists of multiple items/alternatives, each with its own feature vector, and the model must compute utilities for the entire choice set before making a prediction [2]. In other words, the setting refers to a formulation where a sample, {features} → label, might look like:
>
> *One sample = {Alternative A (features), Alternative B (features), Alternative C (features)} → choose one alternative*
>
> while our dataset (BIP) follows a different format, with a distinct prediction task:
>
> *One sample = {Contextual features of a single ride request} → accept or decline*
>
>
> > **Our task has a different data format.**
>
> Each row in our dataset is one observed decision instance with a single contextual feature vector $x_i$, which means all information relevant to the decision (e.g., request time, waiting time, pickup time, surge level, etc.) is stored in one tabular record.
>
> Therefore, there is **no** alternative-level data and **no** choice set per sample. This formulation is widely used in behavioral choice prediction settings [3-5]. In our paper, we project these features (viewed as tokens) to embeddings and advance them by the encoder, after which the aggregated embedding in Eq. (6) is used to predict the choice labels (e.g., whether a driver chooses to accept or decline a request).
>
> > **Why Eq. (6) shows only one feature vector?**
>
> As we described in the paper, Eq. (6) represents the aggregation of contextual representations (i.e., embeddings) of features within **one sample**. Specifically, each sample $i$ contains contextual features, which are embedded and processed through self-attention-based encoder. Eq. (6) then aggregates these feature-level representations into a **single sample-level vector** $h_i$. This single representation is used to predict behavioral choices for the sample (i.e., a decision instance).
> For example, this means that we take the contextual description of one ride request, including features such as request time, waiting time, pickup time, surge level, and driver attributes, and predict whether the driver accepts or declines the request. It is not intended to represent “one vector yields a matrix of per-sample predictions” but rather:
>
>
>
> $x_i \in \mathbb{R}^d
> \xrightarrow{\text{Encoder}}
> H_i^{(L)} \in \mathbb{R}^{d \times d_{\text{hid}}}
> \xrightarrow{\text{Aggregation}}
> h_i \in \mathbb{R}^{d_{\text{hid}}}$
>
>
> This is performed **independently for each sample $i$**, and $\mathbf{h}_i$ is used to estimate behaviors (e.g., accept/decline) for each sample $i$.
>
> We have updated the Related Work section to explicitly clarify the distinction between the traditional DCM setting and our formulation, where each sample corresponds to a single contextual feature vector without alternative-level information.
>
>
> > **Aggregate rather than collapse the set structure entirely.**
>
> According to our problem setting, we need to aggregate the embeddings of contextual features (e.g., request time, waiting time, pickup time, surge level, etc.) for predicting the decision (i.e., accept vs. decline). It is different from the setting you mentioned, with a choice set containing multiple items, each with its own feature vector. Therefore, in our case, the aggregation is important to include all information for prediction, rather than collapse any semantic features such as those for alternatives/items.
>
> [1] Ashkrof, Peyman, et al. "Ride acceptance behaviour of ride-sourcing drivers." Transportation Research Part C: Emerging Technologies 142 (2022): 103783.
>
> [2] Zhang, Shuhan, et al. "Deep Context-Dependent Choice Model." 2nd Workshop on Models of Human Feedback for AI Alignment.
>
> [3] Shahriar, Sakib, et al. "Prediction of EV charging behavior using machine learning." Ieee Access 9 (2021): 111576-111586.
>
> [4] Martín-Baos, José Ángel, et al. "A prediction and behavioural analysis of machine learning methods for modelling travel mode choice." Transportation research part C: emerging technologies 156 (2023): 104318.
>
> [5] Wang, Wenle, et al. "A user purchase behavior prediction method based on xgboost." Electronics 12.9 (2023): 2047.

---

> > ### Author Response · Authors · 2025-11-24
> > **Reply to Reviewer cg7G for Weakenss 2 and Weakness 3**
> >
> > **Weakenss 2: Feature-as-token attention is semantically inappropriate.**
> >
> > **Reply:**
> >
> > We thank the reviewer for the suggestion. To avoid misunderstanding, we clarify again that our task formulation has a totally different setting from what you thought, where each row corresponds to multiple alternatives. Each row in our dataset represents a single behavioral decision instance, not a choice set with multiple items. Therefore:
> > - There are no “items/alternatives” to represent as tokens,
> > - Item-as-token attention is not applicable to our problem, and
> > - We model dependencies across features, not alternatives.
> >
> > > **Why do we use feature-as-token instead of item-as-token?**
> >
> > Our encoder indeed follows the standard tabular-Transformer formulation where each feature is mapped to an embedding and processed as an individual token (Sec. 3.1–3.2). This design is consistent with prior deep tabular models such as TabTransformer. Because there are no alternative-specific features in our data, there are no "alternatives/items" that can be treated as tokens. Therefore, item-as-token attention is not meaningful in this setting.
> >
> > > **QKV projection purpose and feature exchangeability.**
> >
> > The QKV projections in our encoder are designed to learn contextualized interactions among features, such as how pickup time interacts with waiting time or how driver characteristics may influence surge level. These are **feature–feature dependencies** (e.g., how pickup time interacts with waiting time; how rider rating interacts with surge level), not “item–item halo effects.”
> >
> >
> > In addition, **there is no ordering of the features in tabular data**, so it is natural to treat features as exchangeable and allow the attention mechanism to learn their relative importance from data. Here, exchangeability refers to the features within a single decision instance, not to a set of alternatives. Since our dataset does not contain multiple alternatives per row, treating features as exchangeable does not mean that our attention mechanism attempts to model “context or halo effects among items”. Those effects only occur in settings where multiple alternatives coexist within one choice set, which is not the case in our data. Therefore, this setting is fully consistent with our inputs and our modeling objective.
> >
> > > **We do not model dependencies across alternatives; instead, we model dependencies across features.**
> >
> > Instead of capturing halo effects as normally done for items, our model needs to reason about how attributes within the same decision instance interact. It is not meaningful to represent dependencies across multiple alternatives, since we do not have such concepts in our tasks. In other words, what matters are intra-instance feature relationships, such as how pickup time may moderate the influence of surge level, or how waiting time interacts with driver experience. These are feature-level interactions within one sample, rather than cross-alternative dependencies as in choice-set models (e.g., halo effects).
> >
> > As a result, to reflect the influence of features, we analyze the feature importance that is more meaningful insignt for our task. For example, in Fig.2c, we showed the top-10 influential features in the BIP scenario, where only the initial accept/reject decision (Choice 1) is predicted.
> >
> >
> >
> > **Weakness 3: Missing comparisons with relevant sequential choice models.**
> >
> > **Reply:**
> >
> > We thank the reviewer for the suggestion. As noted in our response to your Comment 1, we have clarified the difference between our tasks and the choice modeling in multi-alternative/item setting.
> >
> > The mentioned sequential-choice models assume that at each step the decision-maker faces a set of alternatives/items, each with its own feature vector, and the model compares utilities across these alternatives over time to make selection. **This is not the structure of our dataset.** Taking the AIP setup in our paper as an example, each sample contains one ride request and two binary decisions (e.g., accept/decline), both referring to the same request. There is no alternative-level inputs. Therefore, sequential-choice models based on multi-alternative choice sets are not compatible with our data format.
> >
> > The previous models for similar tasks of ours (i.e., the tabular multi-task prediction) often rely on one-step prediction with respect to single-instance contextual features. Therefore, appropriate baselines in our tasks are tabular models that operate on single-instance contextual features (e.g., TabNet, TabM, CatBoost). We have compared these models in our experiments, and results show that these tabular baselines achieve competitive performance, and our model consistently outperforms them. Taking the AIP setting as an example, TabM achieves F1 of 0.8131 (Choice 2) and 0.6777 (Choice 1), while TabTransformer reaches 0.7522 and 0.5309, respectively. Our model achieves 0.8233 and 0.7120 on these two tasks, exceeding the strongest tabular models (Table 1).

---

> > > ### Author Response · Authors · 2025-11-24
> > > **Reply to Reviewer cg7G for Weakenss 4**
> > >
> > > **Weakenss 4: Lack of novelty.**
> > >
> > > **Reply:**
> > >
> > > We thank the reviewer for this comment. We would like to clarify that PDFormer is not simply a minor variant of TabTransformer. Although we adopt a standard contextual encoder as many tabular models do, this is not the main focus of our work. Importantly, none of them model sequential, dependent choices within the same decision instance, which is the focus of our work.
> > > Our contribution is therefore not just “a variant of TabTransformer with two additional feed-forward heads”, but proposing a progressive dependent-head modeling framework for sequential choices within the same sample - something existing dual-Transformer models do not support. Specifically, the novelty of our work is listed below:
> > >
> > > **(i) Within-instance sequential choice modeling (new problem setting)**
> > >     We model the dependency between Choice 2 and Choice 1, and both occur within the same decision instance. Existing dual Transformers often assume two independent tasks or independent modalities. They provide no mechanism to explicitly represent "How one predicted target influences the next", which is practical for some real-world behavioral choice prediction.
> > >
> > > **(ii) Head-to-head connector**
> > > To address the underrepresentation of dependency, we introduce a learnable connector that projects the first head’s logits into a representation used by the second head, such that:
> > > - We explicitly transfer the outcome of Choice 1 into Choice 2.
> > > - We control dependency through a gated residual mechanism.
> > > - We create a conditional prediction pipeline that is absent from existing model architectures.
> > >
> > > This architectural novelty enables the effectiveness of progressive dual-head structure, which has shown significantly better results than various baselines according to our experiments.
> > >
> > > **(iii) Gated residual dependency control (another innovative design)**
> > > We further design a gated residual that regulates how strongly the later decision depends on the earlier one, while no prior dual Transformer includes dependency-control gates between decision heads. This design prevents over-conditioning, improves training stability, enhances predictive performance in sequential decisions. As shown in the Ablation Study in Section 4.7, we have shown that the gated residual improves the performance in terms of various metrics.
> > >
> > >
> > > **(iv) A unified framework for multi-target dependent prediction in tabular data**
> > > Current tabular Transformers (e.g., TabTransformer, FT-Transformer) cannot explicitly model target dependencies. Our model is the first model to integrate dependency between behavioral choice predictions. It fills a methodological gap in the tabular multi-task prediction, characterized by novel designs:
> > > - The shared contextual encoder that constructs a decision-specific latent state,
> > > - The head-to-head connector that projects and embeds the first decision’s logits into the next decision’s representation,
> > > - The gated residual mechanism that adaptively controls how strongly the first choice influences the second, thus preventing error propagation,
> > > - The conditioned prediction heads that enable explicit dependency modeling while maintaining stability and interpretability.
> > >
> > > These architectural components are not only conceptually novel but also show strong empirical effectiveness, as shown by the following results.
> > >
> > > **(i) Performance lift.** As an early attempt to incorporate progressive dual heads, our approach significantly improves performance by explicitly modeling the dependency between interrelated behavioral choices. Across all datasets, our model generally ranks top, not just on ACC but across AUC, AUCPR, Precision, Recall, F1, MAE, RMSE, R². This consistency indicates the robustness and generalization of our model, rather than just narrow or accidental improvement.
> > >
> > > **(ii) Generalization to multi-choice (n>2) prediction.** We further observe improvements in the new three-choice experiment, showing generalization of our design. We implement the extension from two choices to three-choice prediction by adding a third decision head. The experiments are performed on an additional delivery dataset, which contains three decision targets `is_courier_grabbed`, `is weekend`, and `is prebook`.
> > >
> > > For convenience, we include the results in the below Table 6, Table 7 and Table 8, which again highlight the effectiveness of our approach. Details are provided in Appendix D.

---

> > > > ### Author Response · Authors · 2025-11-24
> > > > **Continue on previous reply for Weakness 4**
> > > >
> > > > Table 6. Model comparison on the delivery dataset for `is_courier_grabbed`.
> > > > | Model| ACC ↑ | AUC ↑ | AUCPR ↑ | Precision ↑| Recall ↑| F1 ↑ |
> > > > |-|-|-|-|-|-|-|
> > > > | Random Forest | 0.8710 (Top-2) | 0.7059| 0.9368| 0.9353 (Top-1)| 0.5149| 0.4944|
> > > > | XGBoost| 0.8671| 0.7404 (Top-2)| 0.9475 (Top-2) | 0.9335 (Top-2)| 0.5000| 0.4644|
> > > > | CatBoost | 0.8671| 0.7090 (Top-3) | 0.9394 (Top-3) | 0.9335 (Top-2)| 0.5000| 0.4644|
> > > > | RealMLP | 0.8671 | 0.5529| 0.8943| 0.8847| 0.8671 (Top-2)| 0.8053 (Top-2)|
> > > > | HyperFast| 0.8703 (Top-3)| 0.7023| 0.9391 | 0.8444 | 0.5150 | 0.4951|
> > > > | TabICL| 0.8721 (Top-1) | 0.5427  | 0.8770 | 0.7969 | 0.5292 (Top-3) | 0.5227 (Top-3)|
> > > > | **3Head-PDHFormer** | 0.8710 (Top-2)| 0.7471 (Top-1) | 0.9504 (Top-1)| 0.8877 (Top-3)| 0.8710 (Top-1) | 0.8147 (Top-1)      |
> > > >
> > > > Table 7. Model comparison on the delivery dataset for `is_weekend`.
> > > > | Model | ACC ↑  | AUC ↑ | AUCPR ↑ | Precision ↑ | Recall ↑ | F1 ↑ |
> > > > |-|-|-|-|-|-|-|
> > > > | Random Forest | 0.9957 | 0.9999| 0.9996 | 0.9965  | 0.9926 | 0.9945   |
> > > > | XGBoost  | 0.9986 (Top-3) | 1.0000 (Top-1) | 1.0000 (Top-1) | 0.9991 (Top-2) | 0.9974| 0.9982 (Top-3) |
> > > > | CatBoost | 0.9985  | 1.0000 (Top-1)| 0.9990| 0.9984| 0.9977 (Top-3)| 0.9981|
> > > > | RealMLP | 0.9833 | 0.9923 | 0.9769| 0.9833| 0.9833| 0.9833|
> > > > | HyperFast | 0.9956  | 0.9999 | 0.9997| 0.9941| 0.9946| 0.9944|
> > > > | TabICL | 0.9995 (Top-1) | 0.9997| 0.9996| 0.9997 (Top-1)| 0.9991 (Top-1)| 0.9994 (Top-1)|
> > > > | **3Head-PDHFormer** | 0.9989 (Top-2)  | 1.0000 (Top-1)| 1.0000 (Top-1)| 0.9989 (Top-3)| 0.9989 (Top-2)| 0.9989 (Top-2)      |
> > > >
> > > > Table 8. Model comparison on the delivery dataset for `is_prebook`.
> > > > | Model | ACC ↑ | AUC ↑ | AUCPR ↑ | Precision ↑ | Recall ↑ | F1 ↑ |
> > > > |-|-|-|-|-|-|-|
> > > > | Random Forest | 0.9645| 0.8613| 0.5032| 0.9822 (Top-2)| 0.5304| 0.5482|
> > > > | XGBoost | 0.9623| 0.9434 (Top-3)| 0.7242 (Top-3)| 0.9811 (Top-3)| 0.5000| 0.4904|
> > > > | CatBoost | 0.9623| 0.9058| 0.5017| 0.9811 (Top-3)| 0.5000| 0.4904|
> > > > | RealMLP | 0.9623| 0.6377| 0.0644| 0.9637| 0.9623 (Top-2)| 0.9437 (Top-2)|
> > > > | HyperFast | 0.9705 (Top-3)| 0.9609 (Top-2)| 0.7173| 0.9763| 0.6113| 0.6740|
> > > > | TabICL | 0.9914 (Top-1)| 0.9561 (Top-3)| 0.8611 (Top-2)| 0.9573| 0.9216 (Top-3)| 0.9387 (Top-3)|
> > > > | **3Head-PDHFormer** | 0.9861 (Top-2)| 0.9874 (Top-1)| 0.8988 (Top-1)| 0.9854 (Top-1)| 0.9861 (Top-1)| 0.9853 (Top-1)|

---

> > > > > ### Author Response · Authors · 2025-11-24
> > > > > **Reply to Reviewer cg7G for Weakenss 5**
> > > > >
> > > > > **Weakness 5. Unclear dataset description and experimental design.**
> > > > >
> > > > > **Reply:**
> > > > >
> > > > > > **Complete and explicit feature descriptions now provided.**
> > > > >
> > > > > We have revised the paper to make the dataset description explicit. Specifically, we now provide a clear summary of all feature groups and representative feature names. This information is included in **Appendix J**.
> > > > >
> > > > >
> > > > > > **Our task is a valid sequential behavioral-choice problem.**
> > > > >
> > > > > As clarified in our response to Comment 1, our task is originally developed from the discrete choice modeling (or choice modeling estimation) in mobility [1], rather than the multi-alternative DCM settings. Our task structure follows the normal form of behavioral choice prediction used in prior work [2-4].
> > > > >
> > > > > We also note that these decisions are **sequential**, since they describe multiple related behavioral outcomes for the same event, rather than independent labels. To make this clearer, we provide a brief explanation of the dataset features and how the decision targets form a sequential setting for each dataset. **We promise all the datasets with implementations will be released upon the paper acceptance**.
> > > > >
> > > > > - The **Urban Mobility Choice Dataset** contains behavioral, temporal, demographic, and contextual features used in the AIP/BIP choice prediction tasks. The complete list of features is shown as below. For example, features such as surge level, waiting time, and pickup distance are known to influence whether a driver accepts or declines a ride request (Choice 1). Under Additional Information Provision (AIP), drivers receive extra attributes which may change their decision in Choice 2.
> > > > >
> > > > >
> > > > >     | Category     | Features |
> > > > >     |-------------|---------|
> > > > >     | All Features | ID, Choice1, Choice2, Req, Time, Time1, Time2, Wait, Dec, Rate, Pickup, Loc, Surge, Long, Cong, Tip, Fare, Block, Workhr, Part, Full, Age, Beginners, Experienced, Acceptance, EarnInc, ExpInc, Satisfied, Taxi, Gender, Partner, Degree, NY, CA, NY_CA, Morning, Midday, Afternoon, Evening, Night, Fac1000, Fac2000, Weekend, Weekend_Friday, Sat_Fri, Sat, Thu_Fri_Sat, Peak_evening, Peak_morning, Peak |
> > > > >
> > > > >
> > > > > - The **new added Delivery Dataset** includes temporal, spatial, and operational features from an online delivery platform. In this dataset, we define the following three binary sequential decision targets: `is_courier_grabbed`, `is_weekend`, and `is_prebook`. For example, the temporal context of the request (is_weekend) often influences whether a delivery can be scheduled in advance (is_prebook).
> > > > >
> > > > >     All features are listed as follows.
> > > > >
> > > > >     | Category     | Features |
> > > > >     |-------------|---------|
> > > > >     | All Features | is_courier_grabbed, is_weekend, is_prebook, sender_lng, sender_lat, recipient_lng, recipient_lat, grab_lng, grab_lat, estimate_arrived_time_hour, estimate_arrived_time_weekday, estimate_arrived_time_minute, dispatch_time_hour, dispatch_time_weekday, dispatch_time_minute, estimate_meal_prepare_time_hour, estimate_meal_prepare_time_weekday, estimate_meal_prepare_time_minute, order_push_time_hour, order_push_time_weekday, order_push_time_minute, platform_order_time_hour, platform_order_time_weekday, platform_order_time_minute |
> > > > >
> > > > >
> > > > > - Another dataset used in the main paper, the **Manufacturing Scenario Sataset**, is derived from an industrial production line. Due to confidentiality, we are currently unable to disclose the full list of feature names.
> > > > >
> > > > >     However, we are actively communicating with the industry partner to release a desensitized version of the dataset, and plan to open-source it once approved.
> > > > >
> > > > >     Meanwhile, all experimental procedures, model configurations, and evaluation protocols are fully documented to ensure scientific transparency.
> > > > >
> > > > >
> > > > > [1] Ashkrof, Peyman, et al. "Ride acceptance behaviour of ride-sourcing drivers." Transportation Research Part C: Emerging Technologies 142 (2022): 103783.
> > > > >
> > > > > [2] Shahriar, Sakib, et al. "Prediction of EV charging behavior using machine learning." Ieee Access 9 (2021): 111576-111586.
> > > > >
> > > > > [3] Martín-Baos, José Ángel, et al. "A prediction and behavioural analysis of machine learning methods for modelling travel mode choice." Transportation research part C: emerging technologies 156 (2023): 104318.
> > > > >
> > > > > [4] Wang, Wenle, et al. "A user purchase behavior prediction method based on xgboost." Electronics 12.9 (2023): 2047.

---

### Official Review · Reviewer_KEok · 2025-10-26

**Soundness:** 4
**Presentation:** 4
**Contribution:** 4
**Rating:** 6
**Confidence:** 3

**Summary:**

The paper introduces PDFormer (Progressive Dual-Head Transformer), a novel deep learning framework for the joint prediction of interdependent behavioral choices. It moves beyond standard parallel Multi-Task Learning by employing a progressive dual-head predictor that explicitly conditions the second choice ($c_2$) on the output of the first ($c_1$) through a head-to-head pathway. The architecture integrates a gated residual mechanism for stability and uses a composite loss with regularization. PDFormer achieves consistent, significant improvements over strong baselines on real-world urban mobility and manufacturing choice prediction tasks, validating the benefit of modeling directional dependency.

**Strengths:**

PDFormer exhibits high originality by introducing a directional multi-task approach, a necessary step beyond parallel-head architectures for truly interdependent choices. The quality of the work is evident in the model's consistent and superior performance across multiple metrics compared to a wide array of strong baselines. The paper's clarity is excellent, featuring a well-motivated design and a clear description of the gated residual mechanism and loss regularization. The inclusion of SHAP analysis further adds to the work's significance by providing valuable interpretability regarding feature importance.

**Weaknesses:**

The work is limited by its strong reliance on knowing the dependency order; an ablation study reversing the task order ($c_2 \rightarrow c_1$) is missing and would strengthen the claim of modeling a directional dependency. The current Dual-Head design restricts immediate application to scenarios with only two interdependent tasks; the paper should discuss the technical challenges and architecture needed for $N>2$ tasks in a dependency chain. Finally, while the manufacturing dataset is a key contribution to model robustness, the paper only presents classification results for the urban mobility data; the full quantitative results for the manufacturing dataset (including R2/PCC for regression targets) should be included in the main text.

**Questions:**

The authors should clarify the technical details of the head-to-head connector: what specific signal (logits, probabilities, or intermediate embedding) is passed from Head 1, and what is the dimensionality of this concatenated input for Head 2? Please include an ablation study on task order to experimentally confirm that the assumed directional dependency is indeed optimal for prediction. For broader applicability, please discuss how the progressive architecture can be extended to a chain of $N>2$ tasks and the associated challenges like error propagation. To fully support the model's claimed robustness, please provide the complete results for the manufacturing dataset in the main paper, including appropriate regression metrics if applicable.

---

> ### Author Response · Authors · 2025-11-24
> **Reply to Reviewer KEok for Weakness 1 and Question 2**
>
> **Weakness 1 and Question  2: The work is limited by its strong reliance on knowing the dependency order; an ablation study reversing the task order is missing and would strengthen the claim of modeling a directional dependency. Please include an ablation study on task order to experimentally confirm that the assumed directional dependency is indeed optimal for prediction.**
>
> **Reply:**
>
> Thank you for the suggestion. We would like to point out that this ablation has already been included in Section 4.7 Ablation Study. We have the ablation variant (iii) “Choice 1 $\rightarrow$ Choice 2”, which reverses the original prediction direction and confirms that the assumed directional dependency (Choice 2 $\rightarrow$ Choice 1) yields the best performance. The results for the reversed task order are presented in Table 5, labeled as 'PDHFormer (Choice 1 $\rightarrow$ Choice 2)' for the AIP scenario and 'PDHFormer (Choice B $\rightarrow$ Choice A)' for the manufacturing scenario.
>
> Table 5: Ablation study for PDHFormer under different scenarios. Choice 1/ Choice 2 and Choice A/ Choice B corresponds to the specified input settings in each scenario. Top-1 results are highlighted in **bold**
>
> **AIP Scenario — Choice 2 (2)& Choice 1 (1)**
> | Model                            | ACC ↑ (2) | AUC ↑ (2) | AUCPR ↑ (2) | Prec ↑ (2) | Recall ↑ (2) | F1 ↑ (2) | ACC ↑ (1) | AUC ↑ (1) | AUCPR ↑ (1) | Prec ↑ (1) | Recall ↑ (1) | F1 ↑ (1) |
> |---------------------------------|------------|------------|---------------|-------------|---------------|------------|------------|------------|---------------|-------------|---------------|------------|
> | PDHFormer (No-GR)               | 0.8410     | 0.7895     | 0.5182        | 0.8138      | 0.8410        | 0.8145     | 0.7168     | 0.7541     | 0.6143        | 0.7021      | 0.7168        | 0.6946     |
> | PDHFormer (Parallel)            | 0.8353     | 0.7867     | 0.5042        | 0.8076      | 0.8353        | 0.8124     | 0.7023     | 0.7452     | 0.5950        | 0.6852      | 0.7023        | 0.6839     |
> | PDHFormer (Choice1→Choice2)     | **0.8497** | 0.7953     | **0.5301**    | **0.8276**  | **0.8497**    | **0.8257** | 0.7052     | 0.7312     | 0.5972        | 0.6907      | 0.7052        | 0.6651     |
> | PDHFormer (Choice2→Choice1)     | 0.8439     | **0.8015** | 0.5257        | 0.8205      | 0.8439        | 0.8233     | **0.7341** | **0.7579** | **0.6261**    | **0.7242**  | **0.7341**    | **0.7120** |
>
> **Manufacturing Scenario — Choice A (A)& Choice B (B)**
> | Model                         | ACC ↑ (A) | AUC ↑ (A) | AUCPR ↑ (A) | Prec ↑ (A) | Recall ↑ (A) | F1 ↑ (A) | ACC ↑ (B) | AUC ↑ (B) | AUCPR ↑ (B) | Prec ↑ (B) | Recall ↑ (B) | F1 ↑ (B) |
> |-------------------------------|------------|------------|-------------|------------|---------------|-----------|------------|------------|-------------|------------|---------------|-----------|
> | PDHFormer (No-GR)             | 0.8606     | 0.9519     | 0.9362      | 0.8605     | 0.8606        | 0.8604    | 0.8355     | 0.9311     | 0.8792      | 0.8328     | 0.8355        | 0.8286    |
> | PDHFormer (Parallel)          | 0.8652     | 0.9547     | 0.9383      | 0.8675     | 0.8652        | 0.8646    | 0.7481     | 0.8784     | 0.8171      | 0.7413     | 0.7481        | 0.7353    |
> | PDHFormer (ChoiceB→ChoiceA)   | **0.9266** | **0.9851** | **0.9803**  | **0.9282** | **0.9266**    | **0.9267**| 0.7658     | 0.8942     | 0.8382      | 0.7627     | 0.7658        | 0.7639    |
> | PDHFormer (ChoiceA→ChoiceB)   | 0.8745     | 0.9593     | 0.9477      | 0.8758     | 0.8745        | 0.8746    | **0.8374** | **0.9322** | **0.8822**  | **0.8342** | **0.8374**    | **0.8307** |

---

> ### Author Response · Authors · 2025-11-24
> **Reply to Reviewer KEok for Weakness 2 and Question 3**
>
> **Weakness 2 and Question 3: The paper should discuss the technical challenges and architecture needed for tasks in a dependency chain. please discuss how the progressive architecture can be extended to a chain of tasks and the associated challenges like error propagation.**
>
> **Reply:**
>
> We thank the reviewer for this helpful comment. Our current presentation focuses on the dual-head case for clarity, but the progressive architecture can naturally be extended to a dependency chain with more than two interdependent tasks. In such setting, we keep the shared contextual encoder unchanged and attach multiple decision heads in sequence.
>
> To further illustrate this extension beyond the two-head case, we also added experiments with three sequential targets on an additional operational delivery dataset. Appendix D (Extended Experiments on an Additional Delivery Dataset) presents a three-head version of our model evaluated on the Meituan dataset (https://github.com/meituan/Meituan-INFORMS-TSL-Research-Challenge) and compared with the strongest baselines from the main paper:
> - In Appendix D.1, we briefly introduce the additional dataset features and prediction targets.
> - In Appendix D.2, we elaborate how the proposed technique can be easily extended from two-choice to three-choice prediction (choice = 3) by adding a third decision head.
> - In Appendix D.3, we report complete experimental results and figures under this extended setting.
> - In Appendix D.4, we compared the model performance with baselines in Table 6, Table 7 and Table 8.
>
> Table 6, Table 7 and Table 8 summarizes the classification performance across all models for the three prediction targets: `is_courier_grabbed`, `is_weekend`, and `is_prebook`.
>
> Table 6. Model comparison on the delivery dataset for `is_courier_grabbed`.
> | Model| ACC ↑ | AUC ↑ | AUCPR ↑ | Precision ↑| Recall ↑| F1 ↑ |
> |-|-|-|-|-|-|-|
> | Random Forest | 0.8710 (Top-2) | 0.7059| 0.9368| 0.9353 (Top-1)| 0.5149| 0.4944|
> | XGBoost| 0.8671| 0.7404 (Top-2)| 0.9475 (Top-2) | 0.9335 (Top-2)| 0.5000| 0.4644|
> | CatBoost | 0.8671| 0.7090 (Top-3) | 0.9394 (Top-3) | 0.9335 (Top-2)| 0.5000| 0.4644|
> | RealMLP | 0.8671 | 0.5529| 0.8943| 0.8847| 0.8671 (Top-2)| 0.8053 (Top-2)|
> | HyperFast| 0.8703 (Top-3)| 0.7023| 0.9391 | 0.8444 | 0.5150 | 0.4951|
> | TabICL| 0.8721 (Top-1) | 0.5427  | 0.8770 | 0.7969 | 0.5292 (Top-3) | 0.5227 (Top-3)|
> | **3Head-PDHFormer** | 0.8710 (Top-2)| 0.7471 (Top-1) | 0.9504 (Top-1)| 0.8877 (Top-3)| 0.8710 (Top-1) | 0.8147 (Top-1)      |
>
> Table 7. Model comparison on the delivery dataset for `is_weekend`.
> | Model | ACC ↑  | AUC ↑ | AUCPR ↑ | Precision ↑ | Recall ↑ | F1 ↑ |
> |-|-|-|-|-|-|-|
> | Random Forest | 0.9957 | 0.9999| 0.9996 | 0.9965  | 0.9926 | 0.9945   |
> | XGBoost  | 0.9986 (Top-3) | 1.0000 (Top-1) | 1.0000 (Top-1) | 0.9991 (Top-2) | 0.9974| 0.9982 (Top-3) |
> | CatBoost | 0.9985  | 1.0000 (Top-1)| 0.9990| 0.9984| 0.9977 (Top-3)| 0.9981|
> | RealMLP | 0.9833 | 0.9923 | 0.9769| 0.9833| 0.9833| 0.9833|
> | HyperFast | 0.9956  | 0.9999 | 0.9997| 0.9941| 0.9946| 0.9944|
> | TabICL | 0.9995 (Top-1) | 0.9997| 0.9996| 0.9997 (Top-1)| 0.9991 (Top-1)| 0.9994 (Top-1)|
> | **3Head-PDHFormer** | 0.9989 (Top-2)  | 1.0000 (Top-1)| 1.0000 (Top-1)| 0.9989 (Top-3)| 0.9989 (Top-2)| 0.9989 (Top-2)      |
>
> Table 8. Model comparison on the delivery dataset for `is_prebook`.
> | Model | ACC ↑ | AUC ↑ | AUCPR ↑ | Precision ↑ | Recall ↑ | F1 ↑ |
> |-|-|-|-|-|-|-|
> | Random Forest | 0.9645| 0.8613| 0.5032| 0.9822 (Top-2)| 0.5304| 0.5482|
> | XGBoost | 0.9623| 0.9434 (Top-3)| 0.7242 (Top-3)| 0.9811 (Top-3)| 0.5000| 0.4904|
> | CatBoost | 0.9623| 0.9058| 0.5017| 0.9811 (Top-3)| 0.5000| 0.4904|
> | RealMLP | 0.9623| 0.6377| 0.0644| 0.9637| 0.9623 (Top-2)| 0.9437 (Top-2)|
> | HyperFast | 0.9705 (Top-3)| 0.9609 (Top-2)| 0.7173| 0.9763| 0.6113| 0.6740|
> | TabICL | 0.9914 (Top-1)| 0.9561 (Top-3)| 0.8611 (Top-2)| 0.9573| 0.9216 (Top-3)| 0.9387 (Top-3)|
> | **3Head-PDHFormer** | 0.9861 (Top-2)| 0.9874 (Top-1)| 0.8988 (Top-1)| 0.9854 (Top-1)| 0.9861 (Top-1)| 0.9853 (Top-1)|
>
> **For the error propagation**
> 1. Our design mitigates this by letting each later head condition on the representation produced by the previous head rather than on its discrete prediction. This has two effects that significantly reduce error propagation:
> - Soft dependency: logits encode uncertainty, giving later heads richer information than a hard 0/1/2 label.
> - End-to-end optimization: conditioning on continuous representations allows gradients to flow across heads, so the dependency structure is learned jointly.
>
> 2. We added a more challenging 3-choice prediction task and observe:
> - Performance gains over top baselines are larger than in the 2-choice setting, especially for the third target.
> - This confirms that modeling inter-choice dependencies becomes increasingly beneficial as n>2.
> - Performance improvement (rather than degradation) shows error propagation is effectively mitigated by our representation-level head-to-head connectors.

---

> > ### Author Response · Authors · 2025-11-24
> > **Reply to Reviewer KEok for Weakness 3 and Question 4**
> >
> > **Weakness 3 and Question 4: Finally, while the manufacturing dataset is a key contribution to model robustness, the paper only presents classification results for the urban mobility data; the full quantitative results for the manufacturing dataset (including R2/PCC for regression targets) should be included in the main text. To fully support the model's claimed robustness, please provide the complete results for the manufacturing dataset in the main paper, including appropriate regression metrics if applicable.**
> >
> > **Reply:**
> >
> > Thank you for the comment. We have moved the complete quantitative results for the manufacturing dataset, including regression metrics (R² and PCC), from the Appendix into the main text as Section 4.6 Results for Manufacturing Scenario: Choice-Linked Regression. The corrosponding results are in main paper Table 4: Model compare for regression prediction on manufacturing dataset
> >
> > Table 4: Model comparison for regression prediction on manufacturing dataset. Top-1/Top-2/Top-3 results are indicated.
> >
> > **Classification Metrics — Choice A**
> >
> > | Model | ACC ↑ | AUC ↑ | AUCPR ↑ | Prec ↑ | Recall ↑ | F1 ↑ |
> > |-------|-------|-------|---------|-------|---------|-------|
> > | Random Forest | 0.8578 | 0.9593 | 0.9492 (Top-3) | 0.8904 (Top-3) | 0.5805 | 0.6067 |
> > | XGBoost | 0.8699 (Top-2) | 0.9566 | 0.9457 | 0.8925 (Top-2) | 0.6217 (Top-3) | 0.6312 (Top-3) |
> > | GBRegressor | - | - | - | - | - | - |
> > | HistGBM | 0.8151 | 0.8377 | 0.7621 | 0.5771 | 0.5793 | 0.5776 |
> > | Decision Tree | 0.8383 | 0.8466 | 0.7745 | 0.6085 | 0.6074 | 0.6080 |
> > | TabNet | 0.8652 (Top-3) | 0.9601 (Top-3) | 0.9442 | 0.8684 | 0.6159 | 0.6171 |
> > | SVM/SVR | 0.8299 | 0.9406 | 0.9196 | 0.8634 | 0.5735 | 0.5835 |
> > | CatBoost | 0.8699 (Top-2) | 0.9632 (Top-2) | 0.9540 (Top-1) | 0.9049 (Top-1) | 0.5969 | 0.6207 |
> > | TabM | 0.7528 | 0.8713 | 0.8055 | 0.7507 | 0.7528 (Top-2) | 0.7456 (Top-2) |
> > | RealMLP | 0.5743 | 0.7889 | 0.7139 | 0.7555 | 0.5743 | 0.4191 |
> > | HyperFast | 0.8559 | 0.9424 | 0.9174 | 0.8635 | 0.5976 | 0.6042 |
> > | TabICL | 0.8225 | 0.9197 | 0.8988 | 0.8464 | 0.5202 | 0.5468 |
> > | **PDHFormer** | 0.8745 (Top-1) | 0.9639 (Top-1) | 0.9536 (Top-2) | 0.8786 | 0.8745 (Top-1) | 0.8752 (Top-1) |
> >
> > **Regression Metrics — Regression A**
> >
> > | Model | MAE ↓ | MSE ↓ | RMSE ↓ | R² ↑ | PCC ↑ |
> > |-------|-------|-------|-------|-----|-----|
> > | Random Forest | 0.0061 | 0.0001 | 0.0075 | 0.7332 | 0.8699 |
> > | XGBoost | 0.0059 | 0.0001 | 0.0072 | 0.7560 (Top-3) | 0.8890 (Top-1) |
> > | GBRegressor | 0.0059 | 0.0001 | 0.0072 | 0.7501 | 0.8780 (Top-3) |
> > | HistGBM | 0.0059 | 0.0001 | 0.0072 | 0.7497 | 0.8780 (Top-3) |
> > | Decision Tree | 0.0056 (Top-2) | 0.0001 | 0.0072 | 0.7543 | 0.8691 |
> > | TabNet | 0.0060 | 0.0001 | 0.0079 | 0.7015 | 0.8378 |
> > | SVM/SVR | 0.0057 (Top-3) | 0.0001 | 0.0071 (Top-1) | 0.7565 (Top-2) | 0.8701 |
> > | CatBoost | 0.0059 | 0.0001 | 0.0072 | 0.7512 | 0.8836 (Top-3) |
> > | TabM | 0.0061 | 0.0001 | 0.0078 | 0.7136 | 0.8456 |
> > | RealMLP | 0.0105 | 0.0002 | 0.0125 | 0.2540 | 0.5145 |
> > | HyperFast | - | - | - | - | - |
> > | TabICL | - | - | - | - | - |
> > | **PDHFormer** | 0.0055 (Top-1) | 0.0001 | 0.0071 (Top-1) | 0.7578 (Top-1) | 0.8751 |

---

> ### Author Response · Authors · 2025-11-24
> **Reply to Reviewer KEok for Question 1**
>
> **Question 1: The authors should clarify the technical details of the head-to-head connector: what specific signal (logits, probabilities, or intermediate embedding) is passed from Head 1, and what is the dimensionality of this concatenated input for Head 2?**
>
> **Reply:**
>
> We thank the reviewer for pointing this out. In response, we have updated Section 3.3 PDHFormer Predictor to include a more detailed description of the **Head-to-Head connector**, clarifying that the logits from Head 1 are linearly projected and concatenated with the aggregated contextual representation for input to Head 2, and specifying the resulting dimensionality as follow:
>
> **Head-to-head connector:**
> To explicitly capture the dependency between the two behavioral choices, the output of the first classification head is incorporated into the second head via a head-to-head connector. Specifically, let $\mathbf{z}\_1 \in \mathbb{R}^{C\_1}$ denote the logits produced by the first head for a sample. We linearly project these logits into a low-dimensional embedding:
>
> $\mathbf{e}\_1 = \mathbf{z}\_1 \mathbf{W}\_c \in \mathbb{R}^{d\_\text{hid}}$
>
>
> where $\mathbf{W}\_c \in \mathbb{R}^{C\_1 \times d\_\text{hid}}$ is a learnable weight matrix. This embedding $\mathbf{e}\_1$ is then concatenated with the aggregated contextual representation $\mathbf{h} \in \mathbb{R}^{d_\text{hid}}$ from the encoder:
>
> $\mathbf{h}\_2 = [\mathbf{h}; \mathbf{e}\_1] \in \mathbb{R}^{2 d\_\text{hid}}$
>
> The concatenated representation $\mathbf{h}\_2$ serves as the input to the second classification head, allowing the prediction of the second choice to be explicitly conditioned on the first. In this way, the model captures interdependencies between the two behavioral choices while preserving the shared contextual information from the encoder.
>
>
> Additionally, we have updated Table 10: Hyperparameters in Appendix F Hyperparameter Settings to report the dimensionality details: the projected embedding from Head 1 is 8-dimensional, and it is concatenated with the 128-dimensional aggregated representation from the PDFormer Encoder to realize the Head-to-Head connector operation, after connection we have the 136-dimension as the Head 2 input.
>
> Table 10: Hyperparameters
> | Category | Details |
> |----------|---------|
> | **Model Hyperparameters** |  |
> | Input Embedding's Input Dimension | Features Nums |
> | Input Embedding Hidden Size | 256 |
> | Input Embedding's Output Dimension | 128 |
> | Encoder Input Dimension | 128 |
> | Encoder Hidden Dimension | 512 |
> | Encoder Output Dimension | 128 |
> | Encoder Block | 2 |
> | Number of Attention Heads | 8 |
> | Head 1 Input Dimension | 128 |
> | Head 1 Hidden Dimension | 32 |
> | Head 1 output Dimension | Class Nums (Choice 2) |
> | Projected Embedding Dimension | 8 |
> | Head to Head Connector Dimension Operation | 128 (Head 1 Input) + 8 (Projected Embedding) |
> | Head 2 Input Dimension | 136 |
> | Head 2 Hidden Dimension | 32 |
> | Head 2 output Dimension | Class Nums (Choice 1) |
> | **Training Hyperparameters** |  |
> | Dropout Rate | 0.4 |
> | Optimizer | AdamW |
> | Learning Rate | 1 × 10^-4 |
> | LR Weight Decay | 1 × 10^-2 |
> | Batch Size | 32 |
> | Epochs | 20 |
> | Classification Loss Functions | CrossEntropyLoss |
> | Classification Loss Weight α | 2 (Choice 2) |
> | Classification Loss Weight β | 1.2 (Choice 1) |
> | Gate Regularization Loss Weight γ | 0.01 |
> | Gate Weight (Embed Stage 1) | 1 |
> | Gate Weight (Classifier) | 1 |
> | Gate Weight (Regressor) | 1 |
> | Gate Initial Bias | 0 |
> | Gate Initial Weight | Xavier uniform |
> | Gate Initial Gain | 1 |
> | Random Seed | 42 |
> | PyTorch Deterministic Mode | True |
> | PyTorch Benchmarking | False |

---

### Official Review · Reviewer_vkmP · 2025-11-01

**Soundness:** 2
**Presentation:** 3
**Contribution:** 2
**Rating:** 4
**Confidence:** 3

**Summary:**

This paper introduces Progressive Dual-Head Transformer (PDFormer), a Transformer-based architecture designed to predict two correlated categorical decision variables. The core idea is to move away from standard multi-task approaches that predict choices in parallel (implicitly assuming independence) and instead adopt a "progressive" two-step prediction. The model first predicts an initial choice (Choice 1). The output representation from this prediction is then explicitly passed via a "head-to-head connector" to a second prediction head, which then predicts the subsequent choice (Choice 2) to capture interdependent decision
patterns. The architecture also incorporates a gated residual mechanism to stabilize training. The authors demonstrate the effectiveness of their model on two real-world datasets, an urban mobility dataset (ride-hailing decisions) and a manufacturing dataset, showing consistent performance gains over a comprehensive set of machine learning and deep learning baselines.

**Strengths:**

1. The paper is well-written and easy to follow. Figure 1 provides a clear and detailed overview of the PDFormer architecture.

2. The experiment design is comprehensive. The empirical evaluation is a strength of this work. The reports of experiment results are clear. The ablation study as well as the test of the reversed prediction order provide strong evidence that modeling the true underlying causal sequence of decisions is crucial.

**Weaknesses:**

1. Novelty is limited. Other papers (e.g., [1-5]) also propose similar Dual-Transformer models. What are the key differences? The paper could be strengthened by more clearly positioning its contribution in the design of novel models.

2. Two datasets are not enough. More datasets could be added.

3. Generalizability issue. The paper focuses exclusively on the case of two sequential choices ($N=2$). The discussion of a longer sequence of choices ($N>2$) would strengthen the paper's contribution to border applications.

4. The improvement compared with baselines is modest. While PDFormer consistently ranks as the top model, the margin of improvement over the best baseline is modest for some key metrics. For instance, in Table 1, the ACC for Choice 2 improves from 0.8410 (TabICL) to 0.8539 (PDFormer).

[1] Dual Vision Transformer

[2] Dual Transformer for Point Cloud Analysis

[3] Dual Transformer Encoder Model for Medical Image Classification

[4] DTSyn: a dual-transformer-based neural network to predict synergistic drug combinations

[5] Analysing the Behaviour of Tree-Based Neural Networks in Regression Tasks

**Questions:**

1. No Reproducibility Statement. As the author guide, “authors are strongly encouraged to include a paragraph-long Reproducibility Statement at the end of the main text (before references) to discuss the efforts that have been made to ensure reproducibility”.

2. Could you provide some analysis on the computational cost (e.g., training time, inference latency) of PDFormer compared to a parallel-head Transformer baseline?

---

> ### Author Response · Authors · 2025-11-24
> **Reply to Reviewer vkmP for Weakness 1**
>
> **Weakness 1: Novelty is limited. Other papers (e.g., [1-5]) also propose similar Dual-Transformer models. What are the key differences? The paper could be strengthened by more clearly positioning its contribution in the design of novel models.**
>
> [1] Dual Vision Transformer
>
> [2] Dual Transformer for Point Cloud Analysis
>
> [3] Dual Transformer Encoder Model for Medical Image Classification
>
> [4] DTSyn: a dual-transformer-based neural network to predict synergistic drug combinations
>
> [5] Analysing the Behaviour of Tree-Based Neural Networks in Regression Tasks
>
> **Reply:**
>
> We thank the reviewer for raising this point. We agree that our contribution needs to be clearly positioned with respect to prior dual-Transformer architectures [1–5]. In the revised manuscript, we have therefore expanded Section 2 (Related Work) to explicitly discuss these works and to contrast their goals and designs with ours.
>
> In general, the cited works use two Transformer modules for independent input modalities or tasks (e.g., RGB + depth, point cloud + image, drug + cell-line) or simply to enhance representational capacity. None of them model sequential, dependent choices within the same decision instance, which is the focus of our work. Our contribution is therefore not just “using two Transformer heads,” but proposing a progressive dependent-head modeling framework for sequential choices within the same sample—something existing dual-Transformer models do not support. Specifically, the novelty of our work is listed below:
>
>   **(i) Within-instance sequential choice modeling (new problem setting)**
> We model the dependency between Choice 2 and Choice 1, and both occur within the same decision instance. Existing dual Transformers often assume two independent tasks or independent modalities. They provide no mechanism to explicitly represent "How one predicted target influences the next", which is pracitcal for some real-world behavioral choice prediction.
>
> **(ii) Head-to-head connector (not found in any Dual-Transformer paper)**
> To address the underrepresentation of dependency, we introduce a learnable connector that projects the first head’s logits into a representation used by the second head, such that:
> - We explicitly transfer the outcome of Choice 1 into Choice 2.
> - We control dependency through a gated residual mechanism.
> - We create a conditional prediction pipeline that is absent from existing model architectures.
>
> This architectural novelty enables the effectiveness of progressive dual-head structure, which has shown significantly better results than various baselines according to our experiments.
>
> **(iii) Gated residual dependency control (another innovative design)**
> We further design a gated residual that regulates how strongly the later decision depends on the earlier one, while no prior dual Transformer includes dependency-control gates between decision heads. This design prevents over-conditioning, improves training stability, enhances predictive performance in sequential decisions. As shown in the ablation study results in main text section 4.7 Ablation Study, we have shown that the gated residual improves the performance in terms of various metrics.
>
>
> **(iv) A unified framework for multi-target dependent prediction in tabular data**
> The cited models were proposed primarily for vision, point cloud, drug synergy, and so on. **None of them focus on tabular data**. Moreover, neither these models nor standard tabular Transformers (e.g., TabTransformer, FT-Transformer) explicitly model target dependencies. Our model is the first model to integrate dependency modeling for behavioral choice prediction. It fills a methodological gap in tabular sequential choice modeling, characterized by novel designs:
> - The shared contextual encoder that constructs a decision-specific latent state,
> - The head-to-head connector that projects and embeds the first decision’s logits into the next decision’s representation,
> - The gated residual mechanism that adaptively controls how strongly the first choice influences the second, thus preventing error propagation,
> - The conditioned prediction heads that enable explicit dependency modeling while maintaining stability and interpretability.

---

> ### Author Response · Authors · 2025-11-24
> **Reply to Reviewer vkmP for Weakness 2**
>
> **Weakness 2: Two datasets are not enough. More datasets could be added.**
>
> **Reply:**
>
> We thank the reviewer for this comment and agree that additional datasets can further strengthen the empirical evaluation. In the revised manuscript, we therefore include an additional operational delivery dataset and report the results in Appendix D (Extended Experiments on an Additional Delivery Dataset). Specifically, we evaluate our model and the strongest baselines from the main paper on the Meituan dataset (https://github.com/meituan/Meituan-INFORMS-TSL-Research-Challenge). This dataset contains three sequential decision targets, providing a more diverse and realistic testing environment than the two datasets in the main paper. These new experiments, together with the original two tasks in mobility and manufacturing, provide stronger evidence for the effectiveness and generality of our model.
>
> Specifically, Table 6, Table 7 and Table 8 summarize the classification performance across all models for the three prediction targets: `is_courier_grabbed`, `is_weekend`, and `is_prebook`. Overall, our model outperforms the baselines across various metrics.
>
> Table 6. Model comparison on the delivery dataset for `is_courier_grabbed`.
> | Model| ACC ↑ | AUC ↑ | AUCPR ↑ | Precision ↑| Recall ↑| F1 ↑ |
> |------------------|--------------------|--------------------|--------------------|---------------------|---------------------|---------------------|
> | Random Forest | 0.8710 (Top-2) | 0.7059| 0.9368| 0.9353 (Top-1)| 0.5149| 0.4944|
> | XGBoost| 0.8671| 0.7404 (Top-2)| 0.9475 (Top-2) | 0.9335 (Top-2)| 0.5000| 0.4644|
> | CatBoost | 0.8671| 0.7090 (Top-3) | 0.9394 (Top-3) | 0.9335 (Top-2)| 0.5000| 0.4644|
> | RealMLP | 0.8671 | 0.5529| 0.8943| 0.8847| 0.8671 (Top-2)| 0.8053 (Top-2)|
> | HyperFast| 0.8703 (Top-3)| 0.7023| 0.9391 | 0.8444 | 0.5150 | 0.4951|
> | TabICL| 0.8721 (Top-1) | 0.5427  | 0.8770 | 0.7969 | 0.5292 (Top-3) | 0.5227 (Top-3)|
> | **3Head-PDHFormer** | 0.8710 (Top-2)| 0.7471 (Top-1) | 0.9504 (Top-1)| 0.8877 (Top-3)| 0.8710 (Top-1) | 0.8147 (Top-1)      |
>
> Table 7. Model comparison on the delivery dataset for `is_weekend`.
> | Model | ACC ↑  | AUC ↑ | AUCPR ↑ | Precision ↑ | Recall ↑ | F1 ↑ |
> |------------------|--------------------|--------------------|--------------------|---------------------|---------------------|---------------------|
> | Random Forest | 0.9957 | 0.9999| 0.9996 | 0.9965  | 0.9926 | 0.9945   |
> | XGBoost  | 0.9986 (Top-3) | 1.0000 (Top-1) | 1.0000 (Top-1) | 0.9991 (Top-2) | 0.9974| 0.9982 (Top-3) |
> | CatBoost | 0.9985  | 1.0000 (Top-1)| 0.9990| 0.9984| 0.9977 (Top-3)| 0.9981|
> | RealMLP | 0.9833 | 0.9923 | 0.9769| 0.9833| 0.9833| 0.9833|
> | HyperFast | 0.9956  | 0.9999 | 0.9997| 0.9941| 0.9946| 0.9944|
> | TabICL | 0.9995 (Top-1) | 0.9997| 0.9996| 0.9997 (Top-1)| 0.9991 (Top-1)| 0.9994 (Top-1)|
> | **3Head-PDHFormer** | 0.9989 (Top-2)  | 1.0000 (Top-1)| 1.0000 (Top-1)| 0.9989 (Top-3)| 0.9989 (Top-2)| 0.9989 (Top-2)      |
>
> Table 8. Model comparison on the delivery dataset for `is_prebook`.
> | Model | ACC ↑ | AUC ↑ | AUCPR ↑ | Precision ↑ | Recall ↑ | F1 ↑ |
> |------------------|--------------------|--------------------|--------------------|---------------------|---------------------|---------------------|
> | Random Forest | 0.9645| 0.8613| 0.5032| 0.9822 (Top-2)| 0.5304| 0.5482|
> | XGBoost | 0.9623| 0.9434 (Top-3)| 0.7242 (Top-3)| 0.9811 (Top-3)| 0.5000| 0.4904|
> | CatBoost | 0.9623| 0.9058| 0.5017| 0.9811 (Top-3)| 0.5000| 0.4904|
> | RealMLP | 0.9623| 0.6377| 0.0644| 0.9637| 0.9623 (Top-2)| 0.9437 (Top-2)|
> | HyperFast | 0.9705 (Top-3)| 0.9609 (Top-2)| 0.7173| 0.9763| 0.6113| 0.6740|
> | TabICL | 0.9914 (Top-1)| 0.9561 (Top-3)| 0.8611 (Top-2)| 0.9573| 0.9216 (Top-3)| 0.9387 (Top-3)|
> | **3Head-PDHFormer** | 0.9861 (Top-2)| 0.9874 (Top-1)| 0.8988 (Top-1)| 0.9854 (Top-1)| 0.9861 (Top-1)| 0.9853 (Top-1)|
>
> We hope these additions sufficiently address the reviewer’s concerns and further strengthen our model’s robustness and extensibility.

---

> ### Author Response · Authors · 2025-11-24
> **Reply to Reviewer vkmP for Weakness 3 and Weakness 4**
>
> **Weakness 3: Generalizability issue. The paper focuses exclusively on the case of two sequential choices (N=2). The discussion of a longer sequence of choices (N>2) would strengthen the paper's contribution to border applications.**
>
> **Reply:**
>
> Following your suggestion, we have conducted the extension from two-choice to three-choice prediction. Detailed settings and results can be found in **Appendix D: Extended Experiments on An Additional Delivery Dataset**. Specifically,
> - In Appendix D.1, we briefly introduce the additional dataset features and prediction targets.
> - In Appendix D.2, we elaborate how the proposed technique can be easily extended from two-choice to three-choice prediction (choice = 3) by adding a third decision head.
> - In Appendix D.3, we report complete experimental results and figures under this extended setting.
> - In Appendix D.4, we compared the model performance with baselines in Table 6, Table 7 and Table 8.
>
> The aforementioned Table 6, Table 7 and Table 8 present the results across all models for the three-choice prediction targets: `is_courier_grabbed`, `is_weekend`, and `is_prebook`. Thses results again verify that our model outperforms the baselines in terms of various metrics. Therefore, our model has the potential to naturally and readily generalize to an arbitrary number of choices/heads, making our model applicable to more complex sequential decision tasks in practice.
>
> **Weakness 4: The improvement compared with baselines is modest. While PDFormer consistently ranks as the top model, the margin of improvement over the best baseline is modest for some key metrics. For instance, in Table 1, the ACC for Choice 2 improves from 0.8410 (TabICL) to 0.8539 (PDFormer).**
>
> **Reply:**
>
> Thank you for pointing out the margin of improvement on ACC for Choice 2. We agree that the absolute ACC gain may appear modest, but we would like to clarify several important points regarding metric interpretation, task difficulty, and broader performance improvements.
>
> **(i) ACC is only one metric; our model achieves substantially larger gains on most key metrics**
>
> While ACC is a useful summary indicator, it is not the most sensitive metric for imbalanced or multi-factor choice modeling. For the same experiment, our model shows notably larger gains on more informative metrics. AUC, AUCPR, Recall, Prec and F1 all show significantly larger improvements than the ACC margin. For example, in Table 1, for the Choice 2 prediction results, our model improves the performance a lot in contrast to TabICL, which represent significant lifts given the difficulty of Choice 2:
> - Precision is improved from 0.4612 (TabICL) to 0.5257 (our model).
> - Recall is improved from 0.6147 (TabICL) to 0.8439 (our model).
> - F1 is improved from 0.6385 (TabICL) to 0.8233 (our model).
>
> These metrics reflect decision quality more reliably than the single ACC metric, particularly under choice imbalance. Besides, in our case, our model improves all major metrics simultaneously, which is rarely achieved by standard models. For example, in Table 3, for the Choice B prduction results on manufacturing dataset:
> - ACC is improved from 0.7314 (TabICL) to 0.8374 (our model).
> - AUC is improved from 0.8486 (TabICL) to 0.9322 (our model).
> - AUCPR is improved from 0.7789 (TabICL) to 0.8822 (our model).
> - Precision is improved from 0.8257 (TabICL) to 0.8342 (our model).
> - Recall is improved from 0.3328 (TabICL) to 0.8374 (our model).
> - F1 is improved from 0.3252 (TabICL) to 0.8307 (our modelmer).
>
> **(ii) The performance gain of our model is consistent across all datasets, settings, and metrics:**
> - Urban Mobility Choice Dataset, including 2-choice prediction tasks for AIP setting (the model predicts *Choice 2 first* and then *Choice 1* within the same decision instance), and BIP setting (in which the model only predicts *Choice 1*)
> - Manufacturing dataset, including 2-choice classification task and classification–regression task
> - Delivery dataset, which we have added for a 3-choice prediction task.
>
>
> In all these practical tasks and datasets, our model generally ranks top, not just on ACC but across AUC, AUCPR, Precision, Recall, F1, MAE, RMSE, R². This consistency indicates the robustness and generalization of our model, rather than just narrow or accidental improvement.

---

> ### Author Response · Authors · 2025-11-24
> **Reply to Reviewer vkmP for Question 1 and Question 2**
>
> **Question 1: No Reproducibility Statement. As the author guide, “authors are strongly encouraged to include a paragraph-long Reproducibility Statement at the end of the main text (before references) to discuss the efforts that have been made to ensure reproducibility”.**
>
> **Reply:**
>
> Thanks for pointing this out. We promise to ensure the reproducibility of our results upon acceptance of the paper. We have added a Reproducibility Statement at the end of the main text, before the references, as suggested.
>
> **Question 2: Could you provide some analysis on the computational cost (e.g., training time, inference latency) of PDFormer compared to a parallel-head Transformer baseline?**
>
> **Reply:**
>
> We appreciate the reviewer’s suggestion and the references provided. However, the cited works [1–5] are all based on vision transformers, point-cloud transformers, or graph-based architectures, which use high-dimensional spatial inputs (e.g., images, voxel grids, feature maps) and rely on computation-heavy components such as multi-scale attention, convolutional tokenizers, or dense spatial embeddings. In contrast, our work focuses on tabular sequential decision modeling, where the model operates directly on low-dimensional feature vectors without spatial structure.
>
> Because the computation graphs, input representations, and parameterization patterns fundamentally differ between these vision/point-cloud models and tabular Transformers, their computational cost (FLOPs, memory usage, training dynamics) is not directly comparable or transferable to our setting. As a result, adopting those architectures as computational baselines would not be meaningful nor technically coherent.
>
> However, following the reviewer’s suggestion to report computational cost, we adapt our architecture by introducing **parallel dual Transformer encoders**, consistent with prior dual-path designs such as cited works [1–5]. It is important to note that most of these works process the input in two parallel Transformer paths and then merge the outputs at the prediction layer, rather than implementing a "parallel head" design. Concretely, in our follow experiment, instead of a single PDFormer encoder used in the main text, we employ two parallel encoders:
>
> - The first encoder processes the embedded input as
> `x_encoded1 = TransformerEncoder₁(x_embed)`
> - The second encoder processes the same input independently as
> `x_encoded2 = TransformerEncoder₂(x_embed)`
> - The outputs are then aggregated by averaging:
> `x_encoded = 0.5 · (x_encoded1 + x_encoded2)`
>
> This aggregated representation is subsequently fed into the PDFormer Predictor. We compare the performance and inference time of the original single-encoder model and the new dual-encoder model below (results filled with our experimental measurements). Our experiment here are conducted for 30 epochs on the newly added Delivery Dataset under the 3-choice prediction setting. The results present in below Table 1, Table 2, Table 3 and Table 4.
>
> Table 1. Comparison Between Single-Encoder and Parallel-Encoder PDFormer
> | Metric / Model Variant | Single-Encoder PDFormer (Ours)|Parallel-Encoder PDFormer|
> |------------------------|------------------------------------|--------------------------------|
> | Training Time | 394.85 s | 460.77 s |
> | Inference Time (per batch)| 1.67 ms | 2.42 ms |
> | Inference Time (per sample)| 0.0260 ms | 0.0378 ms |
>
> Table 2. Performance Comparison for First Classification Head (`is_courier_grabbed`)
> | Metric | Single-Encoder | Parallel-Encoder |
> |--------|----------------|------------------|
> | Accuracy | 87.10% | 87.09% |
> | AUC-ROC | 0.7471 | 0.7429 |
> | AUCPR | 0.9504 | 0.9498 |
> | Precision | 0.8877 | 0.8786 |
> | Recall | 0.8710 | 0.8709 |
> | F1 Score | 0.8147 | 0.8149 |
>
> Table 3. Performance Comparison for Second Classification Head (`is_weekend`)
> | Metric | Single-Encoder | Parallel-Encoder |
> |--------|----------------|------------------|
> | Accuracy | 99.89% | 99.94% |
> | AUC-ROC | 1.0000 | 1.0000 |
> | AUCPR | 1.0000 | 1.0000 |
> | Precision | 0.9989 | 0.9994 |
> | Recall | 0.9989 | 0.9994 |
> | F1 Score | 0.9989 | 0.9994 |
>
> Table 4. Performance Comparison for Third Classification Head (`is_prebook`)
> | Metric | Single-Encoder | Parallel-Encoder |
> |--------|----------------|------------------|
> | Accuracy | 98.61% | 98.52% |
> | AUC-ROC | 0.9874 | 0.9867 |
> | AUCPR | 0.8988 | 0.8826 |
> | Precision | 0.9854 | 0.9844 |
> | Recall | 0.9861 | 0.9852 |
> | F1 Score | 0.9853 | 0.9842 |
>
> As shown, the parallel dual-encoder design substantially increases computational cost while providing negligible performance gain.
> - Performance changes across all heads are minimal (mostly within ±0.1%), indicating robustness to architectural modification.
> - Training and inference times roughly increase by 45%, reflecting the doubling of the Transformer encoder stack.
>
> The computation cost comparison is added in to paper as **Appendix K — Computational Cost Comparison**.

---

### Official Review · Reviewer_XS8A · 2025-11-01

**Soundness:** 3
**Presentation:** 4
**Contribution:** 3
**Rating:** 8
**Confidence:** 4

**Summary:**

This paper studies how to find a transformer-like framework that can predict joint choices where these two decisions are correlated. Existing methods have either not-so-good performance or do not include the interaction between two choices. A new architecture is proposed to fill this gap, and various experiments have shown its efficacy.

**Strengths:**

Very nice written paper with solid results and clear presentation. The identified research question is clear, and the experiments are sufficient.

**Weaknesses:**

- The problem is well-established, and the model architecture is a variant of the parallel version.
- Minor weakness mainly in writing. Please refer to the questions section.

**Questions:**

- The title/name of the transformer makes me think of this paper: https://arxiv.org/pdf/2301.07945, which has the same name but a different model. Perhaps you may consider a slight twist to identify your model better?
- in line 147-160, it might be better to write each component in a consistent manner ("the" and "an" were both used)
- What is the reason to choose GELU activation?
- I also wonder how this framework can be applied to multiple joint choices  (more than 2).

---

> ### Author Response · Authors · 2025-11-24
> **Reply to Reviewer XS8A Weakness 1 and Question 1, 2 and 3**
>
> **Weakness 1: The problem is well-established, and the model architecture is a variant of the parallel version.**
>
> **Reply:**
>
> While our model is similar to general encoder-decoder architecture in a parallel version, we would like to note:
>
> (i) The dependency between behavioral choices is often ignored, causing this problem still not well addressed.
>
> (ii) As one of the first attempt by applying progressive dual heads, our approach significantly improves the performance of parallel versions by explicitly embedding the dependency between behavioral choices.
>
> (iii) On the other hand, our architecture is not a trivial one from parallel  architectures. We introduce several architectural components tailored specifically for dependent tabular decision prediction:
>    - The shared contextual encoder that constructs a decision-specific latent state,
>    - The head-to-head connector that projects and embeds the first decision’s logits into the next decision’s representation,
>    - The gated residual mechanism that adaptively controls how strongly the first choice influences the second, thus preventing error propagation,
>    - The conditioned prediction heads that enable explicit dependency modeling while maintaining stability and interpretability.
>
> **Questions 1: The title/name of the transformer makes me think of this paper: https://arxiv.org/pdf/2301.07945, which has the same name but a different model. Perhaps you may consider a slight twist to identify your model better?**
>
> **Reply:**
>
> We thank the reviewer for pointing this out, we fully agree that      using a distinct name will avoid potential confusion. Therefore, we have renamed our model to **PDHFormer (Progressive Dual-Head Transformer)** throughout the revised paper.
>
> **Questions 2: in line 147-160, it might be better to write each component in a consistent manner ("the" and "an" were both used)**
>
> **Reply:**
>
> We have revised this to ensure consistent phrasing across all components. We now use a uniform structure (starting with “The … layer/module/mechanism”) for each item in the component list.
>
> **Questions 3: What is the reason to choose GELU activation?**
>
> **Reply:**
>
> We chose the GELU activation because it has been shown to provide smoother and more expressive nonlinear transformations than ReLU-type functions, particularly in transformer-based architectures. Its Gaussian-based formulation gives GELU a probabilistic gating behavior: inputs near zero are softly suppressed, while larger inputs are passed through with higher probability. This smooth gating improves gradient flow, stabilizes optimization, and allows the model to preserve informative signals more effectively. (https://arxiv.org/abs/1606.08415).
> GELU also has been widely adopted in transformer based architectures such as ViViT (https://arxiv.org/abs/2103.15691).

---

> ### Author Response · Authors · 2025-11-24
> **Reply to Reviewer  XS8A Question 4**
>
> **Questions 4: I also wonder how this framework can be applied to multiple joint choices (more than 2)**
>
> **Reply:**
>
> We thank the reviewer for the insightful suggestions. Following your comments, we have substantially expanded our experimental evaluation in the revised manuscript.
>
> Additional dataset and longer-choice setting. We added a new section, Appendix D — Extended Experiments on an Additional Delivery Dataset, where we evaluate our model and selected baselines on an operational dataset from a large online delivery platform Meituan, publicly available at: https://github.com/meituan/Meituan-INFORMS-TSL-Research-Challenge. This dataset contains three sequential decision targets, providing a more diverse and realistic testing environment beyond the two datasets in the main paper.
>
> Extension from two-choice to three-choice prediction. We also extend our framework from two-choice to three-choice prediction. Detailed settings and results are provided in Appendix D:
> - In Appendix D.1, we briefly introduce the additional dataset features and prediction targets.
> - In Appendix D.2, we elaborate how the proposed technique can be easily extended from two-choice to three-choice prediction (choice = 3) by adding a third decision head.
> - In Appendix D.3, we report complete experimental results and figures under this extended setting.
> - In Appendix D.4, we compared the model performance with baselines in Table 6, Table 7 and Table 8.
>
> Table 6, Table 7 and Table 8 summarizes the classification performance across all models for the three prediction targets: `is_courier_grabbed`, `is_weekend`, and `is_prebook`. The results again verify the superiority of our model over the baselines.
>
> Table 6. Model comparison on the delivery dataset for `is_courier_grabbed`.
> | Model| ACC ↑ | AUC ↑ | AUCPR ↑ | Precision ↑| Recall ↑| F1 ↑ |
> |------------------|--------------------|--------------------|--------------------|---------------------|---------------------|---------------------|
> | Random Forest | 0.8710 (Top-2) | 0.7059| 0.9368| 0.9353 (Top-1)| 0.5149| 0.4944|
> | XGBoost| 0.8671| 0.7404 (Top-2)| 0.9475 (Top-2) | 0.9335 (Top-2)| 0.5000| 0.4644|
> | CatBoost | 0.8671| 0.7090 (Top-3) | 0.9394 (Top-3) | 0.9335 (Top-2)| 0.5000| 0.4644|
> | RealMLP | 0.8671 | 0.5529| 0.8943| 0.8847| 0.8671 (Top-2)| 0.8053 (Top-2)|
> | HyperFast| 0.8703 (Top-3)| 0.7023| 0.9391 | 0.8444 | 0.5150 | 0.4951|
> | TabICL| 0.8721 (Top-1) | 0.5427  | 0.8770 | 0.7969 | 0.5292 (Top-3) | 0.5227 (Top-3)|
> | **3Head-PDHFormer** | 0.8710 (Top-2)| 0.7471 (Top-1) | 0.9504 (Top-1)| 0.8877 (Top-3)| 0.8710 (Top-1) | 0.8147 (Top-1)      |
>
> Table 7. Model comparison on the delivery dataset for `is_weekend`.
> | Model | ACC ↑  | AUC ↑ | AUCPR ↑ | Precision ↑ | Recall ↑ | F1 ↑ |
> |------------------|--------------------|--------------------|--------------------|---------------------|---------------------|---------------------|
> | Random Forest | 0.9957 | 0.9999| 0.9996 | 0.9965  | 0.9926 | 0.9945   |
> | XGBoost  | 0.9986 (Top-3) | 1.0000 (Top-1) | 1.0000 (Top-1) | 0.9991 (Top-2) | 0.9974| 0.9982 (Top-3) |
> | CatBoost | 0.9985  | 1.0000 (Top-1)| 0.9990| 0.9984| 0.9977 (Top-3)| 0.9981|
> | RealMLP | 0.9833 | 0.9923 | 0.9769| 0.9833| 0.9833| 0.9833|
> | HyperFast | 0.9956  | 0.9999 | 0.9997| 0.9941| 0.9946| 0.9944|
> | TabICL | 0.9995 (Top-1) | 0.9997| 0.9996| 0.9997 (Top-1)| 0.9991 (Top-1)| 0.9994 (Top-1)|
> | **3Head-PDHFormer** | 0.9989 (Top-2)  | 1.0000 (Top-1)| 1.0000 (Top-1)| 0.9989 (Top-3)| 0.9989 (Top-2)| 0.9989 (Top-2)      |
>
> Table 8. Model comparison on the delivery dataset for `is_prebook`.
> | Model | ACC ↑ | AUC ↑ | AUCPR ↑ | Precision ↑ | Recall ↑ | F1 ↑ |
> |------------------|--------------------|--------------------|--------------------|---------------------|---------------------|---------------------|
> | Random Forest | 0.9645| 0.8613| 0.5032| 0.9822 (Top-2)| 0.5304| 0.5482|
> | XGBoost | 0.9623| 0.9434 (Top-3)| 0.7242 (Top-3)| 0.9811 (Top-3)| 0.5000| 0.4904|
> | CatBoost | 0.9623| 0.9058| 0.5017| 0.9811 (Top-3)| 0.5000| 0.4904|
> | RealMLP | 0.9623| 0.6377| 0.0644| 0.9637| 0.9623 (Top-2)| 0.9437 (Top-2)|
> | HyperFast | 0.9705 (Top-3)| 0.9609 (Top-2)| 0.7173| 0.9763| 0.6113| 0.6740|
> | TabICL | 0.9914 (Top-1)| 0.9561 (Top-3)| 0.8611 (Top-2)| 0.9573| 0.9216 (Top-3)| 0.9387 (Top-3)|
> | **3Head-PDHFormer** | 0.9861 (Top-2)| 0.9874 (Top-1)| 0.8988 (Top-1)| 0.9854 (Top-1)| 0.9861 (Top-1)| 0.9853 (Top-1)|
>
> We hope these additions sufficiently address the reviewer’s concerns and further strengthen the evidence for our model’s robustness and extensibility.

---

### Meta-Review · Area_Chair_gVSY · 2025-12-31

**Summary:**

The paper proposes a new transformer architecture for estimating and solving problems with two consecutive choices. The authors note that reviewer cg7G may have used AI to generate their review. Even if it is not, the review has substantial errors to ignore it. Reviewer XS8A is noninformative because they focus only on issues with the title.

The other two reviewers are lukewarm about the paper. They both point out that the proposed architecture is a minor modification of existing transformers for this kind of data; the problem is niche, and the results are only marginally better than those of state-of-the-art solutions and Random Forest-based approaches.

**Reviewer Concerns:**

The reviewer's concerns are primarily about novelty, the interest of the problem and the results. This is a minor paper that is not incorrect but does not advance the field considerably—clearly a B publication.

**Reviewer Scores:**

The scores would mostly be unmoved by the author's reply. The two reviewers who did their job were on the opposite side of lukewarm, but their comments were aligned.

---

### Decision · Program_Chairs · 2026-01-26

Reject